# Deglacial records of terrigenous organic matter accumulation off the Yukon and Amur rivers based on lignin phenols and long-chain *n*-alkanes

Mengli Cao[1], Jens Hefter[1], Ralf Tiedemann[1,2], Lester Lembke-Jene[1], Vera D. Meyer[3], Gesine Mollenhauer[1,2,3]

[1]Alfred-Wegener-Institut, Helmholtz-Zentrum für Polar-und Meeresforschung (AWI), 27570 Bremerhaven, Germany.
[2]Department of Geosciences, University of Bremen, 28359 Bremen, Germany.
[3]MARUM-Center for Marine Environmental Sciences, University of Bremen, 28359 Bremen, Germany.

Corresponding author:
Mengli Cao, mengli.cao@awi.de
Gesine Mollenhauer, gesine.mollenhauer@awi.de

**Abstract:**

Arctic warming and sea level change will lead to widespread permafrost thaw and subsequent mobilization. Sedimentary records of past warming events during the last glacial–interglacial transition can be used to study the conditions under which permafrost mobilization occurs, and which changes in vegetation on land are associated with such warming. The Armur and Yukon rivers discharging into the Okhotsk and Bering Seas, respectively, drain catchments that have been, or remain until today, covered by permafrost. Here we study two marine sediment cores recovered off the mouths of these rivers. We use lignin phenols as biomarkers, which are excellently suited for the reconstruction of terrestrial higher plant vegetation, and compare them with previously published lipid biomarker data.

We find that in the Yukon Basin, vegetation change and wetland expansion began already in the early deglaciation (ED, 14.6–19 ka BP). This timing is different from observed changes in the Okhotsk Sea reflecting input from the Amur Basin, where wetland expansion and vegetation change occurred later in the Preboreal (PB). In the two basins, angiosperm contribution and wetland extent all reached maxima during the PB, both decreasing and stabilizing after the PB. The permafrost of the Amur Basin began to become remobilized in the PB. Retreat of sea-ice coupled with increased sea-surface temperatures in the Bering Sea during the ED might have promoted early permafrost mobilization. In modern Arctic river systems, lignin and *n*-alkanes are transported from land to the ocean via different pathways, i.e., surface runoff *vs.* erosion of deeper deposits, respectively. However, accumulation rates of

lignin phenols and lipids are similar in our records, suggesting that under conditions of rapid sea-level rise and shelf flooding, both types of terrestrial biomarkers are delivered by the same transport pathway. This finding suggests that the fate of terrigenous organic matter in the Arctic differs both on temporal and spatial scales.

## 1. Introduction

Climate warming caused by anthropogenic perturbation affects the Arctic more strongly than other regions of the world. Warming climate induces environmental changes that accelerate degradation of organic matter (OM) stored in permafrost and promote greenhouse gas release (Strauss et al., 2013; Hugelius et al., 2014; Schuur et al., 2015). Permafrost, or permanently frozen ground, is soil, sediment, or rock that remains at or below 0 °C for at least two consecutive years. It occurs both on land and on the continental shelves offshore, and underlies about 22 % of the Earth's land surface (Brown et al., 2002; Wild et al., 2022). Permafrost regions around the world store twice as much carbon as is contained in the atmosphere at present (Hugelius et al., 2014; Friedlingstein et al., 2020). Across the northern circum-polar permafrost regions, the surface permafrost carbon pool (0–3 m depth) amounts to $1035 \pm 150$ Pg (Hugelius et al., 2014). Warming climate may lead to increased mobilization of this carbon pool, while it also affects the type and extent of the vegetation cover, which in turn influences permafrost stability.

During the most recent interval of rapid global warming from the end of the Last Glacial Maximum (LGM) to the early Holocene (~19–11 ka BP), the climate system underwent large-scale change (Clark et al., 2012). Viau et al. (2008) found that the summer temperatures in eastern Beringia (the non-glaciated region between the Eurasian and the Laurentian ice sheet during the Late Pleistocene) during the LGM were approximately 4°C lower than the present and increased rather rapidly toward the Holocene. Sea-ice extent and distribution changed dramatically with consequences for atmospheric moisture content (Ballantyne et al., 2013) and increased heat transport from the oceans to the continental interiors (Lawrence et al., 2008). The increase of precipitation inland as a result of sea ice retreating affects the stability of permafrost in the Arctic (Vaks et al., 2020). Together with increasing air temperatures during the deglaciation, sea-ice retreat may thus have led to rising ground temperatures, active layer deepening and permafrost degradation.

Increased moisture due to sea ice retreat leads to heavier winter snowfall inland (Liu et al., 2012; Park et al., 2013). Menard et al. (1998) reported that groundsurface temperature was higher where ground is covered by thick snow and shrubs and trees than where it was covered by both thin snow and vegetation (moss and lichen). During the last deglacial toward the Holocene, warming ground ice melted, causing the land surface to collapse into space previously occupied by ice wedges, a process called thermokarst. This led to the formation of thermokarst lakes and thermo-erosional valleys as well as rivers, and also likely the release of carbon from thawed deposits (Walter et al., 2006; Walter Anthony et al., 2014). During millennia following the formation of thermokarst lakes, mosses and other plants grew in and around them, which may in part have offset permafrost carbon release (Walter Anthony et al., 2014; Schuur et al., 2015; Turetsky et al., 2020). Several studies suggested major deglacial changes in the vegetation of permafrost-affected areas during the last deglaciation, including the Lena River basin (Tesi et al., 2016), the Yukon Territory (Fritz et al., 2012), the Amur River basin (Seki et al., 2012), and the Sakhalin peninsula and Hokkaido (Igarashi and Zharov, 2011), the latter two bounding the Okhotsk sea to the Northwest and North. Vegetation has a profound impact on distribution and thickness of active layers, and permafrost in cold regions; for example, evidence exists that permafrost temperature in the tundra is lower than in the boreal forest in northwestern Canada (Smith et al., 1998), illustrating the strong effects of vegetation on permafrost stability. Changes in vegetation should therefore be considered when investigating permafrost stability in a changing climate.

Biomarker compositions, distributions, and contents in marine sediments can help to elucidate the vegetation development in adjacent land areas (Winterfeld et al., 2015, 2018; Keskitalo et al., 2017; Martens et al., 2019; Wu et al., 2022). Lignin is a biopolymer exclusively biosynthesized by vascular plants. The relative abundance of individual phenolic monomers varies between different plant types, and the different phenols also differ in their stability towards degradation (Hedges and Mann, 1979; Hedges et al., 1988; Lobbes et al., 2000; Feng et al., 2013; Wild et al., 2022). Ratios between different phenolic monomers are thus sensitive indicators for vegetation type and depositional history of OM originating from land plants (Hedges and Mann, 1979; Hedges et al., 1988).

Previous studies found that the delivery of lignin from land to the ocean is mainly controlled by surface discharge in modern Arctic river systems (Feng et al.,

2013) and has the potential to provide information on surface runoff processes and
wetland extent (Tesi et al., 2016; Feng et al., 2015). Long-chain *n*-alkanes (Alk) with
a strong predominance of the odd carbon number homologues, as well as even-
numbered long-chain *n*-alkanoic acids, derived from the epicuticular waxes of
vascular and aquatic plants (Eglinton and Hamilton, 1967). In contrast to lignin
phenols, sedimentary records of Alk, due to their recalcitrance, likely trace terrigenous
OM which has been mobilized from thawing permafrost deposits in modern Arctic
river systems (Feng et al., 2013) and may be transported into the marine sediment
primarily following coastal erosion during shelf flooding (Winterfeld et al., 2018).
Previous studies have reconstructed the mobilization of terrigenous OM from
degrading permafrost in the Okhotsk (Winterfeld et al., 2018) and Bering shelves
(Meyer et al., 2019) during the last deglaciation based on long-chain *n*-alkyl lipids
results. However, no records exist that combine lignin and Alk data to explore the
potentially different transport of terrestrial OM archived in Arctic marine sediments
during the last deglaciation.
Biomarker records can also be used to infer environmental conditions like sea-
surface temperatures (SSTs) or sea-ice extent that influence heat and moisture
transport from the ocean to the continents. A commonly used proxy for the
reconstruction of SSTs is the $TEX_{86}$, which relies on the relative abundance of so-
called isoprenoid glycerol dialkyl glycerol tetraether lipids (GDGTs) with different
numbers of cyclopentyl moieties (Schouten et al., 2002). These compounds are
derived from the membranes of marine *Thaumarchaeota* and have been found to
record temperature conditions of their habitat. Sea-ice reconstructions rely on the
abundance of highly branched isoprenoids derived from diatoms adapted to life in
sea-ice ($IP_{25}$; Belt et al., 2007). Applying the $TEX_{86}$ temperature proxy, Meyer et al.
(2016) found that atmospheric teleconnections with the north Atlantic were a
widespread control on SST in the northwest Pacific and its marginal seas during the
past 15.5 ka. The $TEX_{86}$-derived SST implies that a widespread of northwest Pacific
SST to atmospheric teleconnections with north Atlantic climate decrease during the
early deglaciation, and summer insolation and $CO_2$ concentration in atmosphere were
important factor driving the SST evolution during this time (Meyer et al., 2016).
GDGTs can also be used to reconstruct terrigenous input to the ocean, when the
relative abundance of branched GDGTs derived mainly from soil bacteria and of
isoprenoid GDGTs is quantified by the branched and isoprenoid tetraether (BIT)

index (Hopmans et al., 2004). According to lignin phenols and the BIT index, Seki et al. (2014a) found that terrestrial OM from the Amur River is a major source of OM in the North Pacific Ocean at present and that terrestrial OM in surface sediments is dominated by gymnosperms in the Okhotsk Sea.

In this study, we present downcore records of lignin phenols from the early deglaciation to the Holocene obtained from sediment cores from off the Amur and Yukon rivers draining permafrost affected by deglacial climate change. These sites in the Ohkotsk and Bering Seas, respectively, record conditions at two contrasting river-dominated continental margins in the North Pacific area. We interpret the lignin phenol records in the context of vegetation and wetland development and investigate the temporal evolution of the different pathways of terrigenous OM export to the ocean by comparing different types of terrigenous biomarker records, i.e., Alk from published studies and new lignin phenol data as well as BIT index values. We further investigate new and published biomarker-based reconstructions of SST, as well as published biomarker-based sea-ice reconstructions to unravel the controls on terrigenous OM transport to the ocean from thawing permafrost landscapes.

## 2. Study area

The Bering Sea is located north of the Pacific Ocean (Figure 1). The Yukon River is the fourth largest river in North America in terms of annual discharge, and drains into the Bering Sea (Holmes et al., 2012). The deglacial sediments from the Bering Sea contain records both of sea-level rise-induced erosion of the vast Bering Shelf, and of runoff from the Yukon River (Kennedy et al., 2010; Meyer et al., 2019). The Yukon Basin was mostly unglaciated during the LGM, but had permafrost (Schirrmeister et al., 2013). Although some permafrost in the Yukon Basin thawed during the last deglaciation (Meyer et al., 2019; Wang et al., 2021), most of the basin is still covered by permafrost today (Fig. 1). Arctic coastal erosion is rapid today, with average rates of erosion at 0.5 m year$^{-1}$ (Lantuit et al., 2012; Irrgang et al., 2022). Sea level rise will lead to greater wave impact on arctic shorelines which increases the coastal erosion (Lantuit et al., 2012). This suggests that during past times of rapid sea-level increase like in the B/A and PB periods coastal erosion was more intense than it is today (Lambeck et al., 2014; Fig. 2, b). Coastal erosion causes a large amount of terrigenous OM to enter the ocean (Couture et al., 2018; Winterfeld et al., 2018), suggesting that during past periods of sea-level rise, similar to today or even stronger erosive forces

were at play supplying vast amounts of terrigenous materials to marine sediments. Pollen records indicate there were no significant changes in vegetation pattern from the LGM to about 16 ka BP, but after about 16 ka BP, birch (one of the first trees to develop after the glacier retreated) pollen became significantly more abundant from western Alaska to the Mackenzie River (Bigelow, 2013). In the early Holocene, significant *Populus-Salix* (cottonwood-willow) woodland development occurred in interior Alaska and in the Yukon Territory (Bigelow, 2013), suggesting both increasing summer temperature and moisture. Alder grows in a warmer and wetter environment than birch, and it is a common genus in Yukon Holocene pollen records (Schweger et al., 2011). Today the catchment of the Yukon River is covered by spruce forest (20 %), grassland (40 %), shrubland (20 %), and open water and wetlands associated with the lowland areas (8 %) (Amon et al., 2012).

The Okhotsk Sea, a semi-enclosed marginal sea located in the west of the North Pacific, is known as the southernmost region of seasonal sea ice in the Northern Hemisphere today (Fig. 1). The continental slope off Sakhalin Island in the Okhotsk Sea receives runoff from the Amur river, the largest river catchment in East Asia. The Amur is also one of the largest rivers in the world in terms of the annual total output of dissolved OM and substantially influences the formation of seasonal sea ice (Nakatsuka et al., 2004). The river originates in the western part of Northeast China and flows east forming the border between China and Russia. The catchment of the Amur transitioned from complete permafrost coverage during the LGM (Vandenberghe et al., 2014) to almost entirely permafrost-free conditions at present. The climate of the Amur Basin today is largely determined by continental patterns from Asia, as the Asia monsoon influences the amount of precipitation from the Pacific transported to this region during the summer. Previous studies found that herbaceous plants were the predominant vegetation in the last glacial periods in the Amur Basin, and were replaced by gymnosperms during the deglaciation and Holocene (Bazarova et al., 2008; Seki et al., 2012). These authors concluded that the variations of vegetation change agree well with climate changes in East Russia with dry conditions in the last glacial followed by wetter climate in the deglaciation and early Holocene (Seki et al., 2012). Now, the vegetation of the Amur Basin belongs mostly to the Taiga zone, with larch as the most common species in the area. The upper reaches of the Amur Basin belong to the coniferous continental taiga at present.

The central areas are dominated by mixed coniferous and broad-leaved forests and the coniferous larch forests are the predominant vegetation in the lower Amur Basin.

## 3. Material and methods

### 3.1. Sampling and age control

Piston core SO202-18-3 (60.13° N, 179.44° W, water depth: 1111 m) and neighboring Kasten core SO202-18-6 (60.13° N, 179.44° W, water depth: 1107 m) were recovered from the northeastern continental slope of the Bering Sea in 2009 during R/V Sonne cruise SO202-INOPEX (Gersonde, 2012). The two cores can be treated as one composite record according to their ultrahigh-resolution micro-X-ray-fluorescence data, sediment facies analysis of laminae and radiocarbon dating results (Kuehn et al., 2014). It represents an apparently continuous sedimentary sequence dated back to the Last Glacial (~25 ka BP) (Kuehn et al., 2014). Selected samples from core SO202-18-6 ($n$ = 20, 10–589 cm core depth, 6.23–12.65 ka BP) and from core SO202-18-3 ($n$ = 29, 447–1423 cm core depth, 12.99–24.1 ka BP) with an average temporal resolution of ~510 years were analyzed for lignin-derived phenol contents.

The 23.7 m-long piston core SO178-13-6 (52.73° N, 144.71° E) was collected from the Sakhalin margin in the Okhotsk Sea during the expedition SO178-KOMEX with R/V Sonne (Dullo et al., 2004) (Fig. 1) with the lowermost interval corresponding to ~17.5 ka (Max et al., 2014). Selected samples ($n$ = 51, 100–2340 cm core depth, 1.11–17.27 ka BP) from core SO178-13-6 were analyzed for lignin-derived phenol contents with an average temporal resolution of ~340 years.

Radiocarbon-based age models for the two cores are from Kuehn et al. (2014) for core SO202-18-3/6 and Lembke-Jene et al. (2017) for core SO178-13-6. The time interval covered by the records will be subdivided into five intervals, the early deglaciation (ED; 19–14.6 ka BP), the B/A (14.6–12.9 ka BP), the Younger Dryas (YD; 12.9–11.5 ka BP), the Pre-Boreal (PB, 11.5–9 ka BP) and the Holocene (＜9 ka BP).

### 3.2. Laboratory analyses

The extraction of lignin phenols was carried out based on the method of Goñi and Montgomery (2000a) and as described in Sun et al. (2017). Dried samples were oxidized with CuO (~500 mg) and ~50 mg ferrous ammonium sulfate in 12.5 ml 2N NaOH under anoxic conditions. The oxidation was conducted with a CEM MARS5 microwave accelerated reaction system at 150 °C for 90 min. After oxidation, known

amounts of recovery standards (ethyl vanillin and trans-cinnamic acid) were added to the oxidation products. The alkaline supernatant was acidified to pH 1 with 37 % HCl. The reaction products were subsequently recovered by two successive extractions with ethyl acetate. The combined ethyl acetate extracts were evaporated under a stream of nitrogen, then re-dissolved in 400 μl pyridine. Prior to injection into the gas chromatograph-mass spectrometer (GC-MS), an aliquot (30 μl) was derivatized with 30 μl bis-trimethylsily-trifluoroacetamide (BSTFA) + 1 % trimethylchlorosilane (TMCS) (60 °C, 30 min). An Agilent 6850 GC coupled to an Agilent 5975C VL MSD quadrupole MS operating in electron impact ionization (70 eV) and full-scan ($m/z$ 50–600) mode was used for analysis. The source temperature of the MS was set to 230 °C and the quadrupole to 150 °C. The GC was equipped with a DB-1 MS column (30 m × 0.25 mm i.d., film thickness 0.25 μm). Helium was used as carrier gas at a constant flow rate of 1.2 ml min$^{-1}$. Samples were injected in splitless mode in a split/splitless injector (S/SL) held at 280 °C. The temperature of the GC-MS column was programmed from 100 °C (initially held for 8 min.), ramped by 4 °C min$^{-1}$ to 220 °C, then by 10 °C min$^{-1}$ to 300°C with a final hold time of 5 min.

Eight lignin-derived phenols were analyzed in this study. They can be classified into three groups according to their plant sources and structures:

1. Vanillyl phenols (V) consisting of vanillin (Vl), acetovanillone (Vn) and vanillic acid (Vd).

2. Syringyl phenols (S), comprising syringealdehyde (Sl), acetosyringone (Sn) and syringic acid (Sd).

3. Cinnamyl phenols (C) that include $p$-coumaric acid ($p$-Cd) and ferulic acid (Fd).

Vanillyl phenols can be found in all vascular plants, syringyl phenols exist only in angiosperms. Cinnamyl phenols are exclusively present in non-woody tissues of vascular plants. Therefore, the S/V and C/V ratios can be used to distinguish lignin between woody and non-woody tissues of angiosperms and gymnosperms (Hedges and Mann, 1979) (Fig. 4). Since S and C phenols are more easily degraded than V phenols during lignin degradation, these two ratios can also be impacted by the degradation degree of lignin (Hedges et al., 1988; Otto and Simpson, 2006) (Fig. 4). Microbial degradation of lignin increase the relative abundance of phenolic acids of V and S phenols, the ratios of vanillic acid to vanillin (Ad/Al)$_v$ and syringic acid to

syringaldehyde (Ad/Al)$_s$ are commonly used to reconstruct the degradation degree of lignin (Ertel and Hedges, 1985; Hedges et al., 1988; Otto and Simpson, 2006) (Fig. 4).

Besides, we also included some other oxidation products that do not necessarily originate from lignin, such as 3,5-dihydroxybenzoic acid (3,5Bd) and para-hydroxybenzenes (P) like $p$-hydroxybenzaldehyde (Pl), $p$-hydroxybenzophenone (Pn), and $p$-hydroxybenzoic acid (Pd). Unlike lignin-derived phenols (V, S, and C), 3,5Bd is absent in plant tissues, but most enriched in peat (Goñi et al., 2000b; Amon et al., 2012). The 3,5Bd/V ratio can be used as a tracer for wetland extent and to determine the degree of degradation for terrigenous OM (Fig. 4).

These compounds were identified based on retention time and mass spectra. Quantification was achieved by peak areas of the respective compounds and using individual 5-point response factor equations obtained from mixtures of commercially available standards analyzed periodically. The yields of Pl, Vl and Sl were corrected by the recovery rate of ethyl vanillin and the recovery rate of trans-cinnamic acid was applied to correct the yield of other lignin-derived compounds and 3,5Bd (Goñi et al., 2000a, b). The standard deviation was determined from repeated measurements of a laboratory internal standard sediment extract ($n$ = 12) and for the carbon-normalized concentration of the sum of the 8 lignin phenols ($\Lambda$8, mg 100mg$^{-1}$ OC) equals 0.31.

Mass accumulation rates (MAR) of vascular plant-derived lignin phenols were calculated as follows:

$$MAR = SR \times \rho, \tag{Eq. 1}$$
$$MAR\text{-}lignin = MAR \times \Sigma8 \div 100 \tag{Eq. 2}$$

where MAR is the mass accumulation rate in g cm$^{-2}$ a$^{-1}$, SR is the sedimentation rate in cm a$^{-1}$, and $\rho$ is the dry bulk density in g cm$^{-3}$. MAR-lignin is the mass accumulation rate of lignin (mg cm$^{-2}$ a$^{-1}$). $\Sigma8$ represents the content of the 8 lignin phenols in mg 10g$^{-1}$ dry sediment.

According to previous studies, the odd-numbered $n$-alkanes in the range of C$_{23}$ to C$_{33}$ are almost exclusively terrigenous (Eglinton and Hamilton, 1967; Otto and Simpson, 2005). Therefore, we can use the mid to long chain length (high molecular weight, HMW) Alk to reflect the contribution of terrigenous OM. Lignin MARs were compared to published MARs of HMW Alk from the Okhotsk Sea (Winterfeld et al., 2018) and the Bering Sea (Meyer et al., 2019). The MARs of HMW Alk from the Bering Sea were recalculated from the published data (Meyer et al., 2019) in order to

compare the results of the two sediment cores. The HMW Alk quantified for the
Bering and Okhotsk Sea sediment cores are $C_{23}$, $C_{25}$, $C_{27}$, $C_{29}$, $C_{31}$ and $C_{33}$ (Fig. 2).

297        Alkanes also have been shown to provide a second marker for wetland extent via

the Paq index (Ficken et al., 2000). It represents the relative proportion of mid-chain
length Alk ($C_{23}$ and $C_{25}$) to long-chain Alk ($C_{29}$ and $C_{31}$). The Paq ratios shown in our
manuscript have been published by others. The Paq ratio of SO202-18-3/6 core was
published by Meyer et al. (2019). The Paq ratio of SO178-13-6 core was published by
Winterfeld et al. (2018). We also cited the Paq ratio of the XP07-C9 core (Seki et al.,
2012), which was retrieved from the Okhotsk Sea (Fig. 1).

304        We further report here the relative abundances of isoprenoid GDGT lipids. These

data were obtained together with the same total lipid extracts that were used for Alk
data published by Meyer et al. (2019). The isoprenoid GDGTs were determined using
the methodology described in Meyer et al. (2016). In brief, the internal standard of
GDGTs ($C_{46}$-GDGT) was added to known amounts of dry sediment, and total lipid
extracts were obtained by ultrasonication with dichloromethane:methanol = 9:1
(vol/vol), 3 times. After extraction and saponification, neutral compounds (including
GDGTs) were recovered with $n$-hexane. Different compound classes were separated
by 1% deactivated $SiO_2$ column chromatography. Polar compounds (including
GDGTs) were eluted with methanol:dichloromethane = 1:1 (vol/vol). Afterward they
were dissolved in hexane:isopropanol = 99:1 (vol/vol) and filtered with a
polytetrafluoroethylene filter (0.45 μm pore size). Samples were brought to a
concentration of 2 μg μl$^{-1}$ prior to GDGT analysis. GDGTs were analyzed by high-
performance liquid chromatography and a single quadrupole mass spectrometer (see
Meyer et al. (2017) for more details).

319        The $TEX_{86}$ index can be used as a SST proxy (Schouten et al., 2002), with the

modified version $TEX_{86}^L$ being applicable in settings where SST is below 15 °C (Kim
et al. 2010). The regional calibration of SST and $TEX_{86}^L$ is based on Seki et al.
(2014b).
$TEX_{86}^L$ = log (GDGT-2 / (GDGT-1+GDGT-2+GDGT-3))             (Eq. 3)
SST = 27.2 × $TEX_{86}^L$ + 21.8                           (Eq. 4)

325        The GDGT-1, GDGT-2, and GDGT-3 are isoprenoid tetraether lipids with 1, 2,

and 3 cyclopentane rings, which were detected by a single quadrupole mass
spectrometer. The MS detector was set for selected-ion monitoring of the following
$(M + H)^+$ ions: m/z 1300.3 (GDGT-1), 1298.3 (GDGT-2), 1296.3 (GDGT-3) (Meyer
et al., 2016). SST is the sea surface temperature in °C.

The BIT index proxy is based on the abundance ratio of branched GDGTs to isoprenoid GDGTs (Hopmans et al., 2004); higher BIT values suggest more contributions from terrestrial soil OM (Weijers et al., 2006; Fig. 3). The BIT index is calculated as BIT = (GDGT-I + GDGT-II + GDGT-III) / (GDGT-I + GDGT-II + GDGT-III + GDGT-IV). GDGT-I, GDGT-II, and GDGT-III refer to the concentration of branched GDGT, GDGT-IV refers to the concentration of isoprenoid GDGT (crenarchaeol). The MS detector was set for selected-ion monitoring of the following $(M + H)^+$ ions: m/z 1022 (GDGT-I), 1036 (GDGT-II), 1050 (GDGT-III), and 1292.3 (GDGT-IV) (Meyer et al., 2016).

## 4. Results

### 4.1. Lignin concentrations and MARs

Lignin phenol concentrations were 0.19–1.43 mg $100\text{mg}^{-1}$ OC (Λ8) or 0.20–1.07 mg
$10\text{g}^{-1}$ sediment (Σ8) in the Bering Sea sediments and 0.32–1.29 mg $100\text{mg}^{-1}$ OC or
0.40–2.16 mg $10\text{g}^{-1}$ sediment in the Okhotsk Sea record. Overall, the MAR of lignin
is lower in the Bering Sea than in the Okhotsk Sea (Fig. 2c, d). During the ED, the
lignin MAR began to increase in the Bering Sea sediment and kept increasing until it
reached a maximum (17.70 $\mu\text{g cm}^{-2}\text{ a}^{-1}$) at the B/A-YD transition (Fig. 2c). After the
B/A, the lignin MAR started to decrease in the Bering Sea until the onset of the YD.
The lignin MAR in the Bering Sea reached a more pronounced but short maximum
(20.61 $\mu\text{g cm}^{-2}\text{ a}^{-1}$) at the YD-PB transition, followed by a decrease to the Holocene.
The lignin MAR in the Okhotsk Sea is more variable than in the Bering Sea record
(Fig. 2d). The lignin MAR shows an initial maximum in the B/A, but reaches a more
pronounced second peak (31.16 $\mu\text{g cm}^{-2}\text{ a}^{-1}$) in the early PB. Similar to the Bering Sea,
the lignin MAR decreased after about 11 ka BP and into the Holocene, however, the
lignin MAR in the Okhotsk Sea sediment featured a rather broad maximum between
B/A and early Holocene.

Deglacial changes in the MARs of HMW Alk have previously been reported for the same cores (Meyer et al., 2019; Winterfeld et al., 2018). The MAR of lignin and HMW Alk changed mostly synchronously in the Bering Sea, but in the Okhotsk Sea, the increase in lignin MAR occurred later than in Alk MAR, and notably also later than in the Bering Sea (Fig. 2c, d). Alk MAR in the Okhotsk Sea featured two similar

maxima in the B/A and during the YD-PB transition, while the lignin MAR maximum in the B/A is less pronounced than that at the YD-PB transition. Lignin MAR are more variable than Alk MARs between 10 and 7.8 ka BP.

**4.2. Sea surface temperature in the Bering Sea (BIT and TEX$_{86}^L$)**

Most BIT values in the Bering Sea are below the commonly assumed threshold value of 0.3 (Fig. 3, Ic), above which SST reconstructions are potentially biased by terrigenous isoGDGTs (Weijers et al., 2006). We are confident that in our study area, marine-derived GDGTs dominate over terrigenous GDGTs, suggesting that TEX$_{86}^L$ is not biased by terrigenous input.

The SST estimates derived from the TEX$_{86}^L$ index are shown in Fig. 3, Ic. The deglacial evolution of the TEX$_{86}^L$-derived SST shows an overall warming, from ~4.5 at 23.4 ka BP to 10.8 °C at 12.0 ka BP. The SST in the Bering Sea remained rather constant during the LGM and the ED. The onset of the B/A is characterized by an abrupt temperature increase of ~2 °C, followed by a decrease at the end of the B/A. At the end of the YD, the SST abruptly increased by ~2 °C, while staying rather constant from 11.5 ka BP to 10 ka BP. At the end of PB, the SST decreased slowly by 1°C from 10.5 to 9.0 ka BP. During the Holocene, the SST ranged between 8.0 °C and 9.7 °C.

**4.3. Lignin source and degradation indicators**

Vegetation development can be assessed using the S/V (angiosperm *vs*. gymnosperm) and C/V ratios (woody tissues *vs*. non-woody tissues) (Hedges and Mann, 1979). The 3,5Bd/V and Paq ratios can be used to indicate the change of wetland extent in the study area (Goñi et al., 2000b; Amon et al., 2012). Similar to (Ad/Al)$_s$ and (Ad/Al)$_v$ ratios, S/V, C/V, and 3,5Bd/V ratios are also affected by degradation processes (Ertel and Hedges, 1985; Hedges et al., 1988; Otto and Simpson, 2006).

The S/V and C/V ratios yielded values of 0.36–0.86 and 0.11–0.46 in the Bering Sea (Fig. 4, Ia, b), while slightly higher ratios of 0.41–0.92 and 0.19–0.70 were obtained in the Okhotsk Sea (Fig. 4, IIa, b). The standard deviations for S/V and C/V are 0.08 and 0.10, respectively. In the Bering Sea, the S/V ratios began to increase from 18 ka BP and kept increasing until it reached a maximum in the transition of the YD to the PB. The change of C/V values was not as obvious as S/V values, but it also reached its maximum at YD-PB transition. Subsequently, S/V and C/V ratios decreased during the Holocene and reached minima at the top of the core.

In the Okhotsk Sea, the S/V values increase slowly from the ED to PB, but the
C/V ratios do not display an obvious increase over the same time, except for a
maximum in C/V ratios around 15 ka BP. Minimum values were found in the ED and
values remained rather low before the YD-PB transition. After reaching maxima in
the PB, S/V and C/V ratios decreased during the Holocene, stabilizing at a higher
level than during the ED.
In the Bering Sea, the 3,5Bd/V ratio ranged from 0.09 to 0.20 (Fig. 4, Id)
(standard deviation: 0.02). From the end of LGM to the YD, the 3,5Bd/V ratio
decreased slowly, but began to increase and reached a small local maximum at the
YD-PB transition. During the Holocene, the 3,5Bd/V ratio decreased again and
reached the lowest values near the top of the core.
The 3,5Bd/V ratios in the core from the Okhotsk Sea range from 0.10 to 0.23
(Fig. 4, IId) (standard deviation: 0.02). The values were rather uniform throughout the
record, with the exception of a maximum during the PB, and the ratio remained rather
stable afterwards.
The $(Ad/Al)_s$ and $(Ad/Al)_v$ ratios ranged from 0.19 to 0.80 (standard deviation:
0.24) and 0.51 to 1.04 (standard deviation: 0.26), respectively, in the Bering Sea (Fig.
4, Ie, f). Maxima in $(Ad/Al)_s$ and $(Ad/Al)_v$ were reached in the Holocene and YD. The
Ad/Al ratio in the Bering Sea showed low values during the PB and increased towards
the early Holocene, when highest values of Ad/Al were obtained.
In the Okhotsk Sea, the $(Ad/Al)_s$ and $(Ad/Al)_v$ ratios are overall similar and
range from 0.30 to 0.79 (standard deviation: 0.24) and from 0.22 to 0.89 (standard
deviation: 0.26), respectively (Fig 4, IIe, f). The $(Ad/Al)_v$ ratio decreased slowly until
10.5 ka BP when the biomarker MARs reached maxima. All minima and maxima in
both indices occurred in the PB. Throughout the rest of the Holocene, Ad/Al values
remained rather constant.

## 5. Discussion

### 5.1. Terrigenous OM mobilization during the last deglaciation

Permafrost remobilization has a strong impact on local topography, vegetation, and
OM fate (Feng et al., 2013; Walter Anthony et al., 2014). We observed distinct MAR
peaks of terrigenous biomarkers in both sediment cores, but the temporal evolution of
MARs and the relative magnitude of change differ between the sites.
In the Bering Sea, lignin MAR began to increase at ~17.5 ka BP, which coincides
with the onset of sea level rise (Fig. 2). Wang et al. (2021) found that the Alaskan
mountain glaciers and Laurentide and Cordilleran ice sheets reached their maximum
extent from 20 ka BP to 16.5 ka BP, suggesting permafrost of the Yukon Basin may
not have begun to be remobilized during this time. The Yukon River discharge did not
increase until 16.5 ka BP (Wang et al., 2021), the terrigenous OM transported by
surface runoff thus may not have increased at ~17.5 ka BP. Keskitalo et al. (2017)
found that the OM flux accumulated on the East Siberian Shelf during the PB-
Holocene transition was high and these OM were characterized by high S/V (0.28–
0.90, mean value is 0.50) and C/V values (0.19–0.60, mean value is 0.35) (Fig. 5).
These authors suggested the Ice Complex Deposit (ICD) as a significant source of
OM in the East Siberian Sea during this time. Previous studies indicated that ICD
samples yield relatively high S/V and C/V ratios ranging from 0.47 to 1.01, and from
0.03 to 0.82, indicating the OM is likely to stem from grass-like material typical of
tundra or steppe biome (Schirrmeister et al., 2013; Tesi et al., 2014; Winterfeld et al.,
2015). The S/V (0.50–0.75, mean value is 0.62) and C/V (0.22–0.36, mean value is
0.28) values of the Bering Sea sediment core from 24 ka BP to 17.5 ka BP are high
(Fig. 4, I), which may indicate that this terrestrial OM is likewise derived from the
ICD. The organic-rich ICD on the coasts of the Bering Sea might have been inundated
or eroded by rising sea level in the ED, which may contribute to the lignin MAR in
the Bering Sea at ~17.5 ka BP.
HMW Alk accumulation in the Bering Shelf started to increase around 16.5 ka
BP (Fig. 2), which coincides with the beginning retreat of Alaskan glaciers (Dyke,
2004). The glacial meltwaters drained through the Yukon River and enhanced fluvial
discharge to the Bering Sea (Wang et al., 2021). The ICD in Alaska and Beringia
might have started to be remobilized at ~16.5 ka BP (Meyer et al., 2019; Wang et al.,
2021), subsequently enhancing the MAR of ICD-derived OM off the Bering Shelf.
Thus, the increased S/V, C/V, and Paq values near 17.5 ka BP (Fig. 4, Ia–c) lend
support to the notion that permafrost of the Yukon Basin may have begun to be
remobilized in the ED.
Retreat of sea ice will increase the SST, and open waters increase the moisture
content of the atmosphere, so the transport of heat from the ocean via atmospheric
pathways to continental interiors increases (Ballantyne et al., 2013; Vaks et al., 2020).
Praetorius et al. (2015) found that SST warming commenced around 16.5 ka BP (core

85JC, Fig. 3, If) in the northern Gulf of Alaska, and Méheust et al. (2018) observed rising SST of the northeast Pacific by ~1.5 °C near 16 ka BP (core SO202-27-6, Fig. 3, Ie) which agrees with our $TEX_{86}^{L}$-SST record (core SO202-18-3/6, Fig. 3, Ic). The same authors reconstructed sea-ice extent based on the $IP_{25}$ proxy to decrease from around 16 ka BP in the Northeast Pacific (Fig. 3, Ie). Jones et al. (2020) reported that the sea ice in the Bering Sea is highly sensitive to small changes in winter insolation and atmospheric $CO_2$. Further evidence for regional climate warming in the hinterland of Alaska is provided by the Brooks Range glacial melting during a time of widespread cooling in the Northern Hemisphere (Dyke, 2004; Wang et al., 2021). Combined evidence from SST and sea-ice reconstructions as well as records of glacial melting thus suggest that the permafrost of the Yukon Basin may have begun to be remobilized at ~16 ka BP.

In the B/A, all biomarker fluxes increased and reached short maxima (Fig. 2c, d). The rate of sea level rise also reached a maximum since the LGM. If Alk had been transported to the ocean primarily through coastal erosion, as is the case with the modern Arctic river transport systems (Feng et al., 2013), then Alk MAR would have been at its maximum, but it was not. Warming may have caused widespread permafrost thaw in the Yukon Basin in the B/A. At this time, SST increased, sea-ice cover decreased (Méheust et al., 2018), and an increase in river discharge was reconstructed (Wang et al., 2021), which may have fostered diatom bloom events (Kuehn et al., 2014; Fig. 2).

Increased S/V, and decreased $(Ad/Al)_{s,v}$ in the B/A, suggesting that the OM deposited in the Bering Sea during the B/A may have been derived from ICD. Similar to our findings, Martens et al. (2019) found relatively high lignin fluxes in the Chukchi Sea during the B/A (Fig. 2e) and showed that lignin deposited during this period was poorly degraded. The authors interpreted this degradation state as permafrost OM from ICD being the dominant source. The relative contribution of ICD and the main pathway of transportation (abrupt thaw or gradual thaw on land) cannot be deduced from our data alone. Further analyses may reveal possible ICD contributions to lignin exported to the marine realm during this interval.

During the YD-PB transition, the Northern Hemisphere experienced an abrupt temperature increase, the maxima of biomarker MAR (Fig. 2c) indicate that the permafrost remobilization in the Yukon Basin reached a peak at this time. The Yukon River discharge increased during the PB (Wang et al., 2021) which can also promote

lignin flux. Evidence for widespread permafrost decomposition and wetland expansion at the same time has been reported from the Bering Sea (Meyer et al., 2019), the Siberian-Arctic (Tesi et al., 2016; Martens et al., 2020), and eastern Beringia (Kaufman et al., 2015). Bering Sea sediments deposited during the time intervals of lignin MAR peaks were laminated (Fig. 2c), indicating increased export productivity and terrigenous OM supply may have promoted anoxic conditions during the YD-PB transition (Kuehn et al., 2014).

The rate of sea level change was lower during the PB than that in the B/A, but the MARs of Alk and lignin reached maxima, and the discharge of the Yukon also increased from the B/A to the PB. Therefore, both coastal erosion and surface runoff may affect the transport of Alk and lignin from land to ocean in the Yukon Basin during the last deglaciation.

Different from the Bering Sea records, lignin MAR did not yet increase in the Okhotsk Sea during the ED (Fig. 2d), except for a short peak at the transition from the ED to the B/A. The discharge of the Amur River was low, but the MAR of lignin increased at the transition from the ED to the B/A when the rate of sea level change was rapid. At the same time, the Alk MAR did not change significantly (Fig. 2). This suggests that both lignin MARs in Okhotsk Sea sediment are affected by coastal erosion during the last deglaciation.

SSTs were higher in the Northeast Pacific and the Bering Sea than in the Northwest Pacific and Okhotsk Sea during the ED, and the $IP_{25}$ value was relatively high in the Okhotsk Sea (Fig. 3, IIc), indicating the sea ice of the Okhotsk Sea did not begin to retreat in the ED (Lo et al., 2018). Caissie et al. (2010) found that the first detectable concentration of alkenones in the Bering Sea sediment at 16.7 ka BP occurred earlier than in the Okhotsk Sea, although the Bering Sea is located further north than the Okhotsk Sea. As a result of prevailing sea-ice cover, the permafrost of the Amur Basin may have remained stable in the ED (Vaks et al., 2013, 2020; Winterfeld et al., 2018; Meyer et al., 2019).

The rate of sea-level change reached a peak during MWP-1A (Fig. 2d) which likely also caused an increase in the rate of coastal erosion. Thus, the increased biomarker MAR during the B/A (Fig. 2) may be attributed largely to coastal erosion. This suggests that both types of biomarkers are supplied via the same erosive process during the B/A, in contrast to findings from the modern-day Arctic.

From the YD to the PB, the Northern Hemisphere experienced an abrupt temperature increase and the SST of the North Pacific increased significantly (Max et al., 2012; Riethdorf et al., 2013; Méheust et al., 2016, 2018; Meyer et al., 2016, 2017). All biomarker MARs in the Okhotsk Sea increased and reached maxima in the YD-PB transition. The permafrost of the Amur Basin may have begun to be remobilized coevally with previously reported periods of stalagmite growth starting after the PB in the south of Siberia, which indicates the decay of permafrost and opening of water conduits into the caves (Vaks et al. 2013, 2020). A pronounced lignin flux maximum occurred during MWP-1B, coinciding with a period of enhanced discharge from the Amur River. This implies that hinterland permafrost thawing played a more important role in the land-ocean OM transport during the later deglaciation, which may have had an impact on the oxygen content and nutrient concentration in the Okhotsk Sea Intermediate Water in the MWP-1B (Lembke-Jene et al., 2017).

MARs decrease drastically after maxima in both the Okhotsk and Bering Seas (Fig. 2c, d). The Amur Basin was completely covered with permafrost during the LGM (Vandenberghe et al., 2014) and almost all of the permafrost was lost until today as a result of permafrost mobilization during the last deglaciation. Thus, the contribution of permafrost OM from the Amur Basin to the marine sediment began to decrease in the early Holocene in agreement with the results of Seki et al. (2012).

In summary, the permafrost of the Amur Basin began to be remobilized in the PB later than in the Yukon Basin. We suggest that this was caused by decreased sea ice or increased SST in the Bering Sea during the ED, while the Okhotsk Sea remained ice-covered. We found that during the last deglaciation, lignin and Alk were supplied from land to the ocean via the same combined processes in the Yukon and Amur Basins, including surface runoff and coastal erosion.

## 5.2. Vegetation changes in the two river basins

### 5.2.1. Yukon River Basin

As the climate warmed during the transition from the LGM to the ED, moisture increased and an increasing number of thermokarst lakes developed in Alaska, especially after about 16–14 ka BP (Bigelow, 2013; Walter et al., 2007). We observe an increase in S/V ratios from the ED to the B/A, indicating increasing contributions of angiosperms around this time, extending into the B/A (Fig. 4, Ia). The S/V and C/V ratios are also influenced by the degradation of lignin, with increasing ratios suggesting a lower degradation state (Hedges et al., 1988; Otto and Simpson, 2006),

but there is no parallel decrease in the more commonly used degradation indicator, i.e., the Ad/Al ratios, at the same time (Fig. 4, Ie, f). As the permafrost had begun to be remobilized in the Yukon Basin during the ED, the S/V suggests that the cover of angiosperms in this basin increased.

During the transition of ED-B/A, the rate of sea level change was rapid, implying that the distance of our study site to the river mouth must have changed. The degree of OM degradation, however, did not change significantly at the same time (Fig. 4, e, f). Reports that the degradation degree of Lena River-derived OM and surface sediments of Buor Khaya Bay are similar (Winterfeld et al., 2015) suggest that the oxidative degradation of lignin occurred mainly on land, in agreement with our results. We thus posit that transport within the ocean that might have increased in distance and duration in response to the sea level change may only have a limited impact on OM degradation in the Bering Sea. The S/V and C/V can thus be regarded to mainly reflect vegetation development in the transition of ED-B/A. Anderson et al. (2003) also found birch pollen becoming significantly more prevalent after about 16 ka BP from western Alaska to the Mackenzie River, suggesting that these regions were characterized by glacier retreating and more favorable climatic conditions. Coevally, the Paq index (Fig. 4, Ic) shows an increase, indicating wetland expansion.

There is evidence that herb-dominated tundra was replaced by *Betula-Salix* (birch-willow) shrub tundra around Trout Lake (the northern Yukon) during the B/A (Fritz et al., 2012) as the climate warmed and became more humid than during the ED. In line with these observations, our C/V ratios indicate that the contribution of non-woody plant tissues was lower in the B/A than in the ED (Fig. 4, Ib).

After the B/A, the summer temperatures during the YD dropped by ~1.5 °C (Fritz et al., 2012), thus cold-adapted non-arboreal plants briefly increased in abundance (Fritz et al., 2012). The S/V ratios indicate that the non-woody angiosperm plants' contribution reached a maximum in the Yukon Basin during the YD-PB transition (Fig. 4, Ia). The MARs of biomarkers in the Bering Sea also reached maxima during this transition (Fig. 2, c). Since the opening of the Bering Strait (~11 ka BP, Jakobsson et al., 2017), a trend of increase in *Betula* (birch) was observed in eastern Beringia (Fritz et al., 2012; Kaufman et al., 2015) which indicates a progressively more maritime climate developing in response to changes in the marine environment (Igarashi and Zharov, 2011). Vegetation development and permafrost remobilization both contribute to the biomarker MARs (Fig. 2, c). The BIT values

higher than 0.3 from 13 ka BP to 10.5 ka BP (Fig. 3, Ic) further support this interpretation while at the same time indicating that in these intervals, $TEX_{86}^L$ cannot be used to reflect the sea surface temperature change during this interval. Additionally, the intensification of oxygen minimum zones in the Bering Sea during the B/A (Fig. 2c) may be related to the increase in surface runoff (freshwater and OM fluxes; Kuehn et al., 2014).

Since vegetation responds to changes in both temperature and moisture, significant *Populus/Salix* (cottonwood/willow) woodland development occurred in interior Alaska and the Yukon Territory during the early Holocene (Anderson et al., 2003). However, the expansion of these angiosperm plants is not reflected in our S/V record (Fig. 4, Ia), the interpretation of S/V ratios may be complicated by the influence of degradation processes during the early Holocene. Pollen assemblages from northern Siberian soils have shown that woody plants occurred only after the onset of the Holocene (Binney et al., 2009), which agrees with a decrease in the C/V ratios since the early PB into the early Holocene (Fig. 4, Ib).

We compare our S/V and C/V ratios with published values from sediment cores, surface sediments and suspended materials in the Arctic and subarctic (Fig. 5). Such plots can help to identify the main types of plant tissues the lignin phenols are derived from, and enable the detection of potential degradation effects.

The S/V and C/V ratios from our Bering Sea core compare favorably with those from a core recovered from the Chukchi shelf covering parts of the B/A and the YD as well as the late Holocene (Martens et al., 2019) (Fig. 5). This may suggest that a similar type of vegetation prevailed across much of Beringia. After the opening of the Bering Strait, Pacific waters flowed into the Chukchi Sea, and it is conceivable that the terrestrial material transported to the Bering Sea by the Yukon River may have been in part transported into the Chukchi Sea. The top of our core dates to the early Holocene, a period that was characterized by more widespread broad-leaf angiosperm vegetation than today, which might explain the offset between our early Holocene S/V and C/V ratios and those reported for Yukon River surface sediment (S/V: 0.28, C/V: 0.14) (Feng et al., 2015) and dissolved organic carbon in the Yukon (S/V: 0.47, C/V: 0.14) (Amon et al., 2012) at present. Degradation of lignin in sediments may explain some of the discrepancies between sediment data and S/V and C/V ratios reported from suspended materials collected in the modern Lena River (Fig. 5). As the Amur River catchment is dominated by gymnosperms at present (Seki et al. 2014a), the S/V

ratios of the Amur River and Okhotsk Sea surface sediments (Seki et al., 2014a) are lower than in the Bering Sea core (Fig. 5).

The highest values of the 3,5Bd/V ratio correlate with the enhanced degradation of lignin phenols around 17.5 ka BP (Fig. 4, Id). This suggests that degraded OM is the dominant source of the lignin phenols at this time, in agreement with previous studies (Meyer et al., 2016, 2019). Global melt water pulses according to Lambeck et al. (2014) occurred during the following periods: MWP-1A from 14.6 to 14.0 ka BP and MWP-1B from 11.5 to 10.5 ka BP. The 3,5Bd/V and $(Ad/Al)_s$ ratios decreased slowly from the ED to the MWP-1A, which indicates that the change in 3,5Bd/V values from the LGM to the early B/A reflects a variable degree of OM degradation, rather than expansion of wetlands or peatlands. The 3,5Bd/V also featured a short maximum during the late YD and early PB when the 3,5Bd/V signal is likely dominantly ascribed to increases in wetland or peatland sources, as there is no parallel maximum in Ad/Al ratios (Fig. 4, Id, e, f).

This pattern of 3,5 Bd/V change is not in agreement with the Paq ratio determined for the same core earlier (Meyer et al., 2019), although both proxies may reflect wetland expansion (Goñi et al., 2000b; Amon et al., 2012). The temporal evolution of Paq is similar to that of S/V and C/V, where Paq began to increase in the ED and reached its maximum in the YD-PB transition (Fig. 4, Ic), indicating that the proxies are influenced to some extent by degradation.

The Ad/Al ratios decreased when the biomarker MAR peaked during MWP-1B (Fig. 4, IIe, f) which may correspond to better preservation during rapid burial, or higher contribution of ICD OM and fresh angiosperm debris. This better preservation, is in agreement with previous studies (Anderson et al., 2003; Meyer et al., 2019). Kuehn et al. (2014) found that increases in biological export production and remineralization of OM in the Bering Sea during the B/A and PB reduced oxygen concentration to below 0.1 ml $L^{-1}$ and caused the occurrence of laminated sediments (Fig. 2c). This anoxic condition in the Bering Sea during the B/A and PB also slowed down rates of OM decomposition and increased the accumulation of OM.

In summary, our data together with published evidence indicate that in the Yukon Basin, vegetation change and wetland expansion began in the ED. Angiosperm plant contribution and wetland extent all reached their maxima during the PB, both decreasing and stabilizing at lower levels after the PB. During the PB, terrigenous

OM appeared least degraded, suggesting rapid supply and burial of rather well-preserved terrigenous OM.

**5.1.2. Amur River Basin**

The lowest contribution of non-woody angiosperms as indicated by low S/V and C/V ratios occurred at 16.6 ka BP (Fig. 4, IIa, b). Subsequently, both ratios increased and reached maxima during the PB, suggesting the expansion of angiosperms and non-woody tissues contributing substantially to lignin. After the PB, the S/V decreased rapidly and remained stable during the Holocene, while the C/V ratio showed a second maximum at 9.2 ka BP suggesting an increasing contribution of non-woody angiosperms in the PB (Fig. 4, IIa, b). In agreement with our data, published lipid records provide evidence that the vegetation in the Amur Basin did not change significantly in the ED (Seki et al., 2012). Winterfeld et al. (2018) found a general synchronicity of Amur River discharge and the northward extent of monsoon precipitation in the early Holocene. Climate warming associated with high moisture supply allowed the expansion of birch-alder forests in the Amur Basin in the PB (Bazarova et al., 2008; Igarashi and Zharov, 2011).

C/V and S/V ratios indicate that the contribution of non-woody gymnosperm tissue was higher in the early than in the later deglaciation (Fig. 6), similar to what has been reported from East Siberian Shelf records (Keskitalo et a., 2017). The YD caused only minor vegetation changes in the East Asian hinterland (Igarashi and Zharov, 2011). Our lignin records for this period are in agreement with previous studies that indicated that the Lower Amur River basin mainly featured shrub birch-alder forests and rare *Pinus* (Bazarova et al., 2008; Seki et al., 2012).

Non-woody angiosperm plant contributions to the Okhotsk Sea sediment strongly increased during the PB (Fig. 4, IIa, b), when the summer insolation and regional temperatures reached the highest values since the LGM. Significant vegetation changes in the Amur Basin thus started in the PB period, temporally offset from the Yukon Basin, and the contribution of angiosperms from 14.6 ka BP to 9 ka BP appears to be higher than during the ED (Fig. 6). Bazarova et al. (2008) reported based on pollen analyses that a turning point in vegetation development in the Amur Basin occurred at a boundary of 10 ka BP. The Middle Amur depression registered the first appearance of broad-leaved species of pollen and a prevalence of spores over arboreal pollen at that time (Bazarova et al., 2008). The C/V ratio did not decrease as rapidly as the S/V ratio after the peak and showed a second maximum at ~9 ka BP.

Some pollen of *Picea* (such as *P.* glauca and *P.* mariana) yield exceptionally high amounts of cinnamyl phenols (Hu et al., 1999), which may have affected the C/V ratio as the end member of woody/non-woody tissues. An et al. (2000) concluded that lakes were deepest and most extensive around 10 ka BP in northeastern China (the upper Amur basin), and 3,5Bd/V and Paq values reached maxima at the same time (Fig. 4, IId, c) suggesting wetland extent peaked during the PB. Therefore, wetland plants that have broad leaves, such as sedges, may also have a positive influence on the C/V ratio.

The S/V and C/V data from the Holocene part of our core do not agree with published values for the Okhotsk Sea and Amur River surface sediments (Fig. 6). During the past 250 years, vegetation was marked by significant rises of gymnosperms, such as pines, combined with the reduction in the swamp area and a large increase in fire activity (Yu et al., 2017), likely resulting in higher contributions of gymnosperm to the surface sediment while these changes are not resolved in the samples analyzed for our record.

The 3,5Bd/V and Paq ratios of the Okhotsk Sea both display relatively high values during the PB (Fig. 4, IId, c). Seki et al. (2012) found high Paq values during the PB in a nearby sediment core XP07-C9, and the values in their core were higher than in ours (Fig. 4, IIc). Spores of Sphagnum show a distinct peak during the PB (Morley et al., 1991), reflecting an expansion of mesic and boggy habitats. Our records together with published evidence thus suggest that permafrost destabilization and wetland expansion in the Amur Basin occurred only at the beginning of the PB, while those processes were initiated much earlier in the Yukon basin.

The Ad/Al values were decreasing until 10.5 ka BP and reached minima during the PB (Fig. 4e, f), indicating that low temperatures on land on the one hand, and rapid burial in marine sediments during shelf flooding and coastal erosion during MWP-1B on the other hand, contributed to the Ad/Al signals. The 1 ka averages of the S/V, C/V ratios show similar minima as the Ad/Al ratios from the ED to the PB (Fig. 4II), suggesting that degradation processes exert a strong control on the S/V and C/V ratios during a time when vegetation did not change in the Amur Basin. In the course of climate amelioration from around 11.6 ka BP (Tarasov et al, 2009), the rates of vegetation development, wetland expansion and Amur River discharge (Fig. 2f) all displayed maxima in the PB. Generally, higher Ad/Al values in the later part of the PB suggest that fluvial runoff supplied more degraded lignin. Aerobic degradation of OM in soils by fungi has also been shown to increase Ad/Al values (Goñi et al., 1993).

Since the oxidative degradation occurred mainly on land (Winterfeld et al., 2015), and lateral transport is likely short, this increased degradation is unlikely to occur in the ocean. The Okhotsk Sea shelf is narrower than the Bering Shelf and Siberian shelves, the lateral shelf transport times (i.e., the cumulative time a particle spends in sedimentation-resuspension cycles) of the Okhotsk shelf are therefore likely to be much shorter than what has been reported for the Laptev Shelf (Bröder et al., 2018), further supporting our interpretation.

The rate of sea level change in the Bering Sea (Manley, 2002) is slower than the global average rate (Lambeck et al., 2014). The effect of sea level change on the degradation process of terrestrial OM in the Bering Sea is limited. We are not aware of a published local reconstruction of sea level change for the Okhotsk Sea from 20 ka BP to the present, but we suggest that, since the shelf of the Okhotsk Sea is narrower than that of the Bering Sea, the effect of sea level change on the Okhotsk shelf may be neglected. Pre-aged OM and young OM can be transported from land to the marine sediments in a variety of ways, such as coastal erosion and surface runoff, but the relative contribution of different carbon pools could not be quantified by lignin and Alk fluxes or other parameters (S/V, C/V, 3,5Bd/V, Ad/Al, and Paq), as they appear to be transported through the same pathways during the last deglaciation. Further investigation using compound-specific radiocarbon analysis is needed to quantify the contribution of different carbon pools in marine sediments.

In summary, our records indicate that in the Amur Basin vegetation change and wetland expansion began during the PB and in the early Holocene, in agreement with previous paleo-vegetation studies. This timing is different from observed changes in the Yukon Basin. However, similar to the Yukon Basin, the wetland extent and non-woody angiosperm contribution were reduced and stabilized after the PB in the Amur Basin. The increased vegetation and wetland indices, as well as increased degradation of lignin in the Okhotsk Sea sediment at the end of the PB, may be related to changes in the source of OM (shelf and coastal erosion *vs*. river transported material).

**Conclusions:**

By analyzing mass accumulation rates of terrigenous biomarkers in sediments from the Bering and Okhotsk Seas, we provide the first downcore records of lignin from the Yukon and Amur Basins covering the early deglaciation to the Holocene. We find that vegetation changed earlier in the Yukon than in the Amur Basin. Although S/V, C/V and 3,5Bd/V ratios can reflect vegetation change and wetland development, the

degradation state of lignin strongly overprints these proxy signals and should be considered as a function of temperature, transport distance and burial rate. Similar to changes in vegetation, we observe that degradation and remobilization of permafrost of the Yukon Basin also occurred earlier than in the Amur Basin. Sea-ice extent and SSTs of adjacent ocean areas might have had a strong influence on the timing of hinterland permafrost mobilization. Our study reveals that lignin transported by surface runoff may account for significant proportions of lignin during inland warming, but the export of lignin and lipids do not always occur via different pathways, as both biomarker groups can be contributed from rapidly eroding deep deposits during phases of rapid permafrost thaw. In contrast to modern day evidence suggesting different pathways for lipid and lignin biomarker transport, our records imply that during glacial peaks of permafrost decomposition, lipids and lignin might have been delivered to the ocean by identical processes, i.e., runoff and erosion.

**Authors' contributions**

MC measured and compiled lignin data, and wrote the manuscript with the help of all co-authors. JH was responsible for all biomarker analyses. LLJ and RT provided samples. VM carried out sea surface temperature measurements of SO202-18-3/6. GM designed the study. All authors participated in the discussion of results and conclusions and contributed to the final version of the paper.

**Competing interests**
The authors declare that they have no conflict of interest.

**Acknowledgments**

We thank the masters and crews of R/V Sonne for their professional support during cruises SO202 (INOPEX) and SO178 (KOMEX). Hartmut Kuehn is acknowledged for providing total organic carbon and dry bulk density data of site SO202-18-3/6. Mengli Cao thanks the China Scholarship Council (CSC) and POLMAR- Helmholtz Graduate School for Polar and Marine Research for additional support. We are also grateful to the laboratory and computer staff at Alfred Wegener Institute. The biomarker data generated in this study are accessible at the database Pangaea: https://doi.org/10.1594/PANGAEA.948376.

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

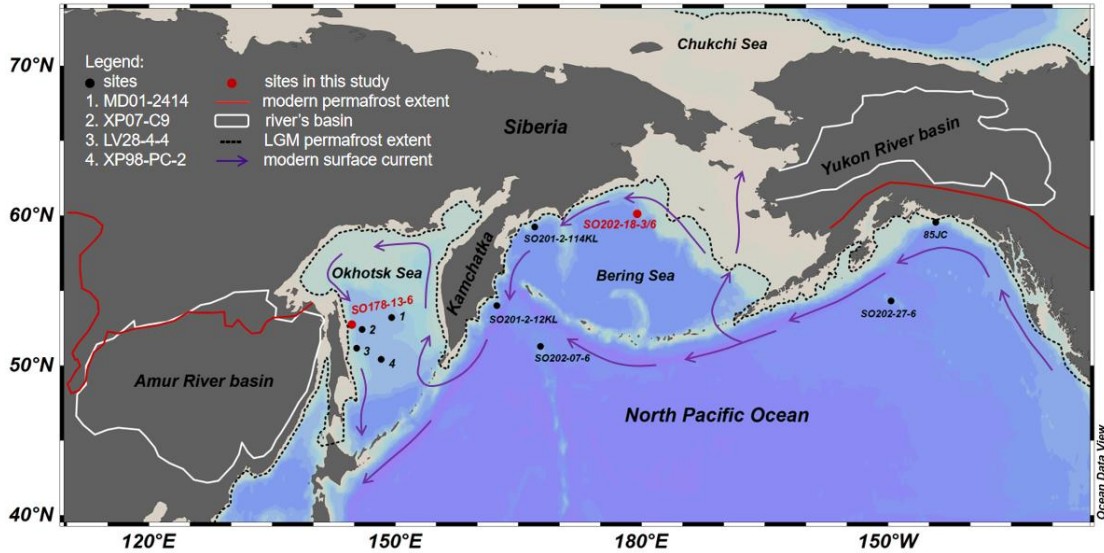

**Figure 1.** Study area. Red dots indicate locations of sediment cores investigated in this study;
black dots denote cores described in previous studies. 1: site MD01-2414 (Lattaud et al., 2019;
Lo et al., 2018). 2: site XP07-C9 (Seki et al., 2012). 3: site LV28-4-4 (Winterfeld et al., 2018).
4: site XP98-PC-2 (Seki et al., 2014b).

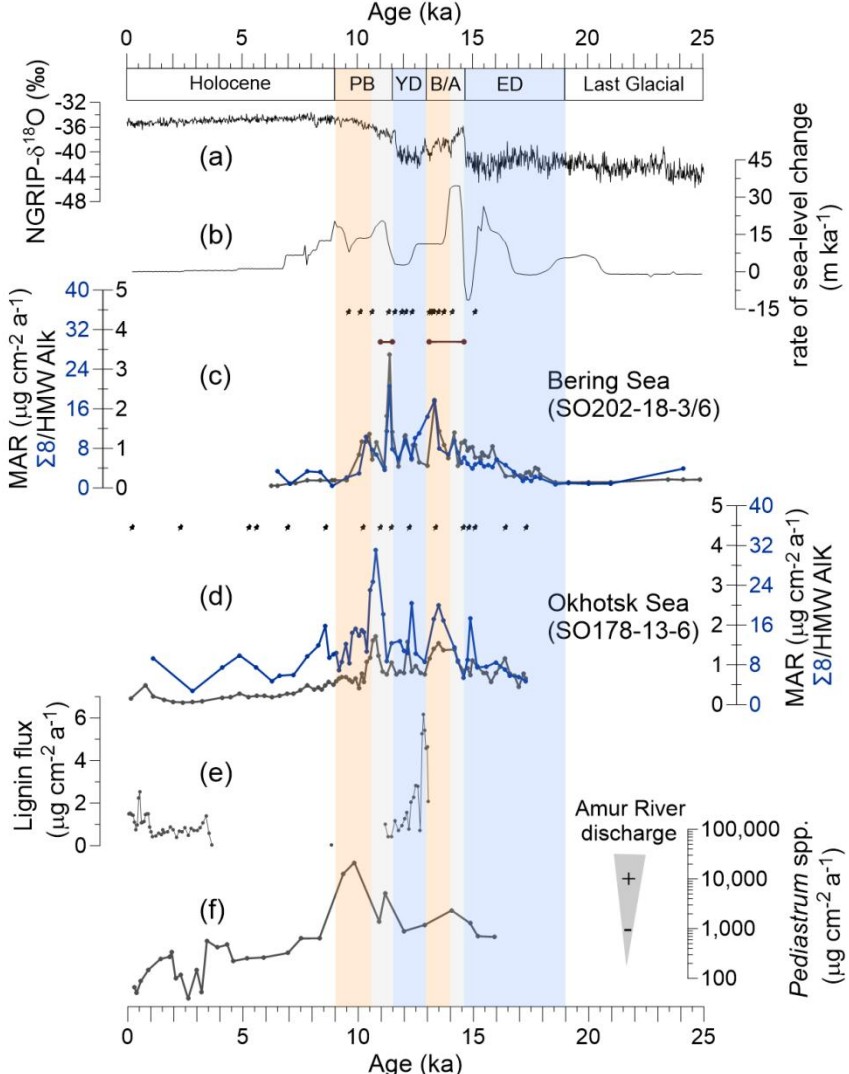

**Figure 2.** Proxy records of terrestrial organic matter supply and environmental records of deglacial changes. (a) Greenland NGRIP $\delta^{18}O$ (Rasmussen et al., 2008). (b) Global rate of sea-level change (Lambeck et al., 2014). (c) MAR of lignin phenols (blue) and HMW Alk (black; Meyer et al., 2019) from core SO202-18-3/6. (d) MAR of lignin phenols (blue) and HMW Alk (black; Winterfeld et al., 2018) from core SO178-13-6. Pin marks at the top of (c) and (d) show age control points, the accelerator mass spectrometry $^{14}C$ dates for SO202-18-3/6 (Kuehn et al., 2014) and SO178-13-6 (Max et al., 2012). Brown bars in panel c indicate laminated/layered (anoxic) core sections (Kuehn et al., 2014). (e) Lignin flux from core 4-PC1 (Chukchi Sea, Martens et al., 2019). (f) Accumulation rate of chlorophycean freshwater algae *Pediastrum* spp. from core LV28-4-4 (Winterfeld et al., 2018). Blue boxes represent the cold spells the early deglaciation (ED) and Younger Dryas (YD), orange boxes are for the warm phases Bølling-Allerød (B/A) and Pre-Boreal (PB). Gray boxes highlight the periods of melt water pulse 1A (MWP-1A) and 1B (MWP-1B).

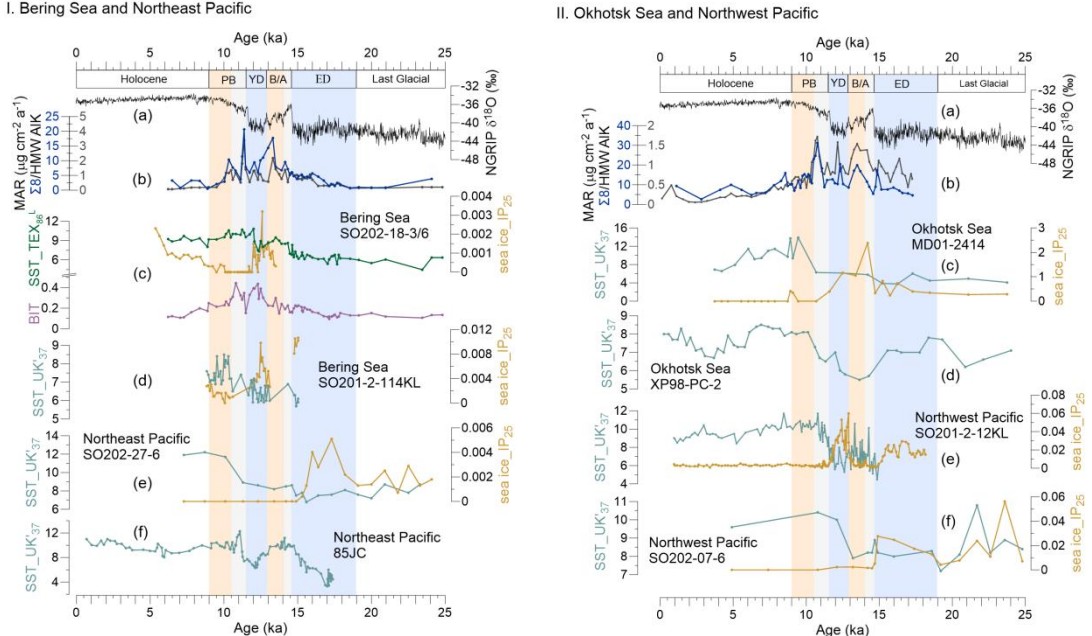

**Figure 3.** Records of sea surface temperature (SST) and sea ice (IP$_{25}$) in the Bering Sea and Northeast Pacific, and Okhotsk Sea and Northwest Pacific during the past 25 ka. (a) NGRIP-$\delta^{18}$O from Greenland (Rasmussen et al., 2008). (b) MAR of biomarkers.

I: (c) The green line reflects SST (TEX$_{86}^L$), and the purple line shows the BIT from this study, SO202-18-3/6. The orange line denotes the IP$_{25}$ obtained for this core by Méheust et al. (2018). (d) SST and IP$_{25}$ for core SO202-2-114KL (Max et al., 2012; Méheust et al., 2016). (e) SST and IP$_{25}$ for core SO202-27-6 in the Northeast Pacific (Méheust et al., 2018). (f) SST for core 85JC (Praetorius et al., 2015).

II: (c) SST and IP$_{25}$ of the core MD01-2414 in the Okhotsk Sea (Lattaud et al., 2019; Lo et al., 2018). (d) SST for core XP98-PC-2 (Seki et al., 2014b). (e) SST and IP$_{25}$ for core SO201-2-12KL in the Northwest Pacific (Max et al., 2012; Méheust et al., 2016). (f) SST and IP$_{25}$ for core SO202-07-6 (Méheust et al., 2018). The units of the SST and IP$_{25}$ are °C and μg g$^{-1}$ sediment, respectively. Blue boxes represent intervals with prevailing colder climate conditions during the early deglaciation (ED) and Younger Dryas (YD), orange boxes are for the warm phases Bølling-Allerød (B/A) and Preboreal (PB). Gray boxes highlight the periods of melt water pulse 1A (MWP-1A) and 1B (MWP-1B).

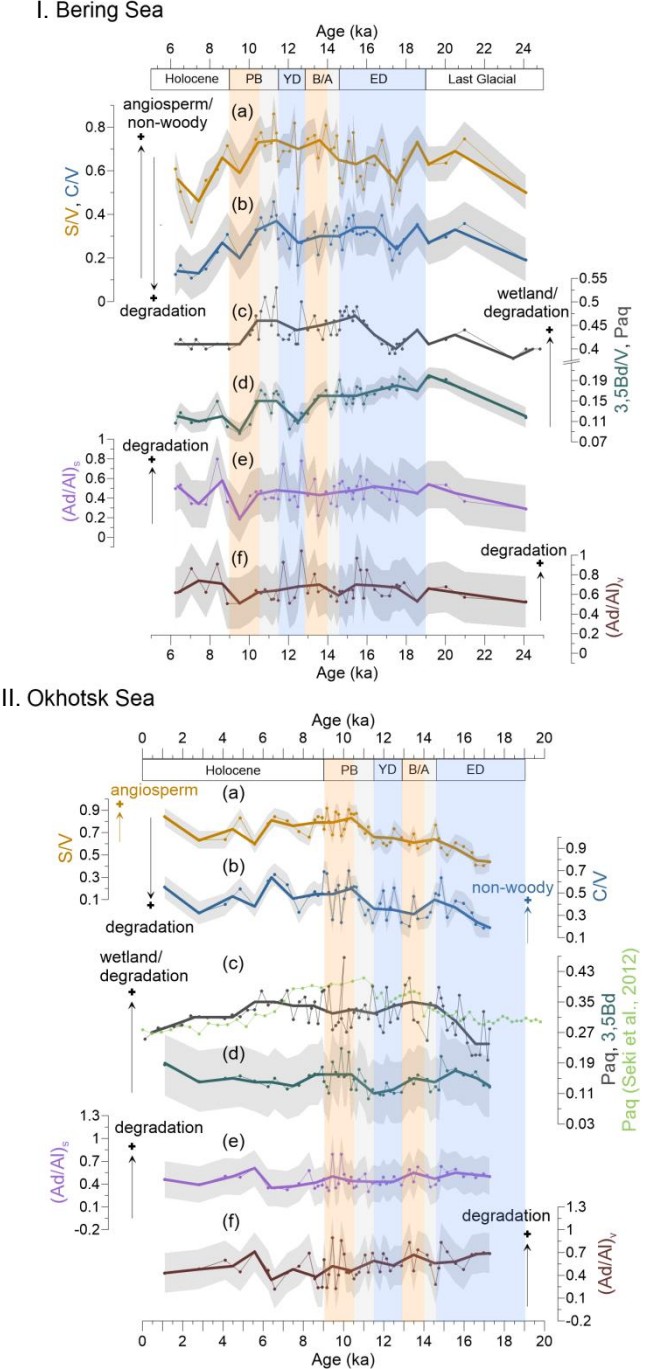

**Figure 4.** Records of lignin and non-lingin phenol indices compared with the Paq index in Bering and Okhotsk Sea sediments. (a and b): S/V and C/V ratios reflect the vegetation change and/or degree of lignin degradation in the respective river basins. (c and d): 3,5Bd/V and Paq ratios represent the wetland extent or degree of degradation in the respective catchments. In panel II showing records from the Okhotsk Sea, the light green line represents the Paq of a nearby core, XP07-C9 (Seki et al., 2012). (e and f): The Ad/Al can reflect the degradation of lignin phenols. Grey shaded areas illustrate the uncertainty of these indices. Bold lines are the 1 ka averages of the corresponding indices. Blue boxes represent the cold spells the early deglaciation (ED) and Younger Dryas (YD), orange boxes are for the warm phases Bølling-Allerød (B/A) and Pre-Boreal (PB). Gray boxes highlight the periods of melt water pulse 1A (MWP-1A) and 1B (MWP-1B).

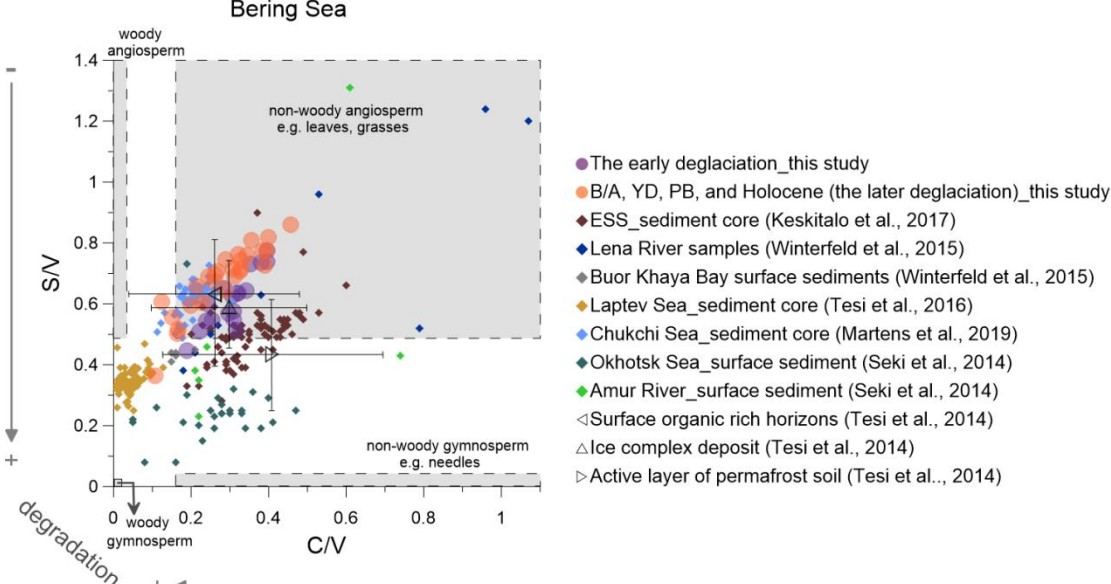

**Figure 5.** Lignin indicators of terrigenous material in the Bering sediment (solid circles) compared with previously studied (Martens et al., 2019; Keskitalo et al., 2017; Tesi et al., 2016; Seki et al., 2014a; Winterfeld et al., 2015). The early deglaciation is from 19 to 14.6 ka BP and after the early deglaciation is the later deglaciation. The dark triangles represent the ratio of S/V and C/V from surface soils, Ice Complex deposits and active layer permafrost (Tesi et al., 2014). ESS is short for the East Siberian Shelf.

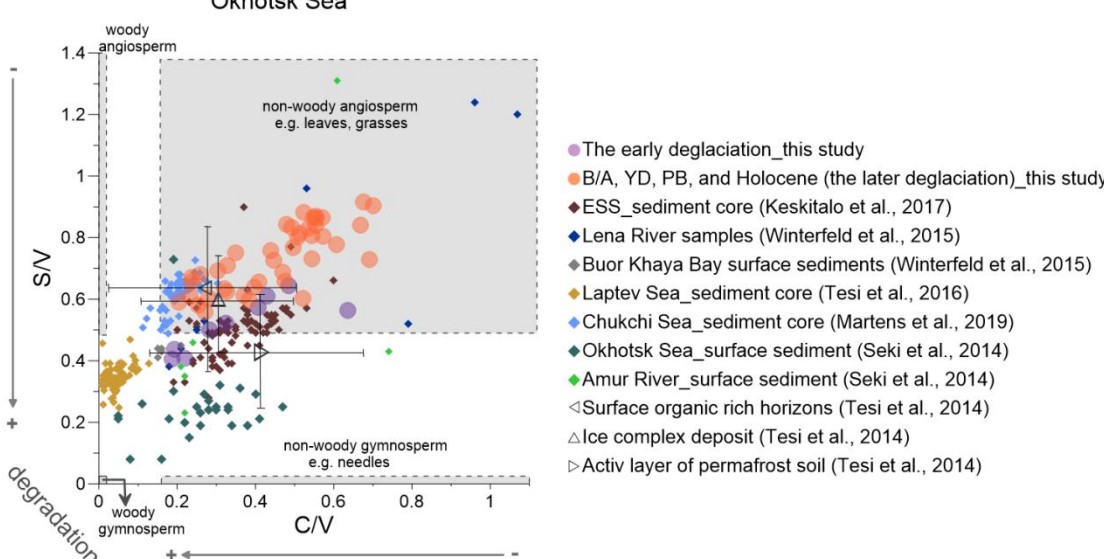

Figure 6. Lignin indicators of terrigenous material in the Okhotsk Sea sediment (solid circles) compared with previously studied (Martens et al., 2019; Keskitalo et al., 2017; Tesi et al., 2016; Seki et al., 2014a; Winterfeld et al., 2015). The early deglaciation is from 19 to 14.6 ka BP and after the early deglaciation is the later deglaciation. The dark triangles represent the ratio of S/V and C/V from surface soils, Ice Complex deposits and active layer permafrost (Tesi et al., 2014). ESS is short for the East Siberian Shelf.