# Peer review of "Deglacial records of terrigenous organic matter accumulation off the"

_Climate of the Past, 2022_

## Referee Comment (RC3)

*Review of Cao et al. "Deglacial records of terrigenous organic matter accumulation off the Yukon and Amur rivers based on lignin phenols and long-chain n-alkanes" (cp-2022-67 )*

Synopsis

The primary focus of this study is to reconstruct vegetation changes in the Okhotsk and Bearing Seas (near the Amur and Yukon Rivers, respectively) since the end of the last glacial period. To do so, the authors report lignin concentrations and molecular compositions from two marine cores, and they compare these to previously published *n*-alkyl lipid records from nearby cores. The authors conclude that vegetation change initiated sooner in the Yukon Basin than in the Amur Basin, and that wetland extent stabilized after the Preboreal. Importantly, the authors also conclude that lignin and *n*-alkyl lipids experience similar accumulation rates at this time, implying a similar transport pathway and delivery mechanism, which contrasts with results on export pathways in modern Arctic river systems.

Overall, I find that this study is interesting and provides a high-quality dataset of lignin concentrations, molecular ratios, and accumulation rates. The results will be of interest to those studying Arctic systems, permafrost mobilization/loss, and terrestrial biomarkers. That said, I concur with the other two reviewers that there is a bit of confusion when reading this manuscript as to what is new vs. what is compiled from previous studies. I highlight this and other general comments below, noting that I do not belabor the points that are already well-articulated in the other two reviews. Once these issues have been sorted, then I fully support this publication of this manuscript in *Climate of the Past*. Please do not hesitate to contact me with any questions regarding this review.

Sincerely,

Jordon Hemingway
jordon.hemingway@erdw.ethz.ch

**General comments:**

L38: It is worth stating in the abstract what is the consequence of "both types of terrestrial biomarkers [being] delivered by the same transport pathway." This contrasts with the modern river system; introducing this contrast (and a short sentence on proposed reasons why) would fit nicely at the end of the abstract.

L51: Add comma after "Holocene"

L76: Of course, the "predominance of the odd carbon number homologues" in $n$-alkyl lipids is only true for alkanes---alcohols and alkanoic acids exhibit the opposite preference!

L79-85: The authors should clarify that this discussion on the difference in transport pathways between alkanes and lignin refers specifically to the modern systems---which they contrast with their own results in the discussion.

L87: I second reviewer 1's opinion that the BIT index should be removed here. As far as I can tell, it is never mentioned again and, given the huge complexity and uncertainty in using this as a terrestrial OC source indicator, this brief mention only raises more questions than it answers.

Sec. 1: Overall, I agree with both previous reviewers (esp. articulated by reviewer 2) that the introduction should be reworded to clearly articulate what is new to this study and what is derived from the literature. This should also mention all of the ratios and metrics that will be used throughout this study, and briefly state their interpretation (e.g., Paq is currently not introduced as a proxy for wetland expansion until Sec. 3.2, and the interpretation of various lignin ratios is currently not clear until the discussion!)

L105: LGM not yet defined.

L187-190: The inclusion of "some other oxidation products that do not necessarily originate from lignin" appears rather out-of-the-blue here, but these compounds are discussed later on. I therefore suggest at least mentioning these compounds and their utility in the introduction so that a reader knows to expect them.

L202-205: I strongly suggest re-writing these "equations" to be proper equations, not just words written in pseudo-equation form. Something like:

"…calculated as follows:

$$MAR = SR \times \rho, \tag{Eq. 1}$$

where MAR is the mass accumulation rate in g cm$^{-2}$ a$^{-1}$, SR is the sedimentation rate in cm a$^{-1}$, and $\rho$ is the dry bulk density in g cm$^{-3}$." Etc. This would greatly simplify the reader's ability to interpret these calculations. Further down, the Paq and TEX "equations" should be restructured similarly.

L212: Again, I would mention Paq in the introduction since this comes a bit out-of-the-blue here. This would all be clarified with a re-write of the introduction to more clearly articulate what is new and what is taken from the literature.

L215: I concur with the comments of reviewer 2 for this section.

L250: Should this be "11.3 ka"?

L275-283: It's not clear to the reader at this point what the "3,5Bd/V" ratio represents or why it is important. As mentioned above, these types of ratios should be first introduced (including their utility and interpretation) in the introduction.

L284-294: My above comment also applies for Ad/Al ratios. These should be mentioned earlier so the reader knows how to interpret, e.g., a range of "0.19 to 0.80".

L296: Are $TEX_{86}$-derived SSTs *really* reliable to one $100^{th}$ of a degree? I suggest adjusting reported precision and honestly reporting $TEX_{86}$ uncertainty here.

Sec. 5.1-5.2: I concur with reviewer 1's suggestion to restructure these sections to begin with a discussion on terrestrial OM sources, fluxes, accumulation rates, etc. and *then* move into an interpretation of these biomarker ratios, including potential impacts of degradation.

L555-556: This is an interesting finding, but it is only mentioned in passing here. Why do the authors think these delivery mechanisms have changed relative to the modern? I would appreciate a bit more discussion on this topic, as I think some reviewers will find it highly relevant.

---

## Author Comment (AC1)

Dear reviewer,

Re: Manuscript ID: cp-2022-67 and Title: Deglacial records of terrigenous organic matter accumulation off the Yukon and Amur rivers based on lignin phenols and long-chain *n*-alkanes. by Mengli Cao et al., Clim. Past Discuss., https://doi.org/10.5194/cp-2022-67-RC1, 2022

We thank you for your precious comments and suggestions concerning our manuscript. Those comments are all valuable and very helpful for revising and improving our study, as well as the important guiding significance to our researches. We have studied comments carefully and have made correction which we hope meet with approval.

**General comments:**

*1. In general, the introduction can be elaborated on, as it doesn't cover all used ratios at the moment. It should fine introduce the $TEX_{86}$ ratio, and why a temperature reconstruction is needed to evaluate the sources of OM into marine sediments. Also, the S/V, C/V, 3,5Bd/V and Ad/Al ratios need to be introduced. How are they generally interpreted, and what records exist from the Arctic already?*

Response: We agree with the reviewer's comment regarding the introduction of our manuscript and realize the need to re-organize the text and include additional aspects. In the revised manuscript, we re-organized the introduction and believe that it now better explains why (biomarker-based) reconstructions of sea-surface temperatures and sea-ice extent are needed to obtain a better understanding of the deglacial changes in permafrost stability and vegetation development in the region. We will also include more detailed descriptions of the proxies used and what they can indicate ($TEX_{86}^{L}$, BIT, S/V, C/V, and so on), but we believe that the right place for these detailed descriptions is the methods section.

*2. As a second suggestion, I would restructure the discussion, starting with section 5.2. on the sources of terrigenous organic matter, that discusses the provenance of OM based on the fluxes. Following up on this, the impact on degradation on section 5.1. vegetation proxies can be discussed, potentially explaining a lot of the short-term variation that is especially apparent in the Yukon sediments. This way, the reader will leave with a well supported interpretation of the vegetation changes, which is the main aim of this manuscript. Then, both the changes in provenance and vegetation can be compared with the temperature and sea ice proxies.*

Response: We thank the reviewer for this constructive suggestion. We started with vegetation change in the discussion section because that is the main thrust of this study. However, we noticed that following the comments given by the reviewer may make it easier for readers to understand our main points step by step. Therefore, we will change the structure of the discussion section, discussing organic matter sources based on biomarker concentrations first, followed by vegetation development in the two basins.

**In-line suggestions:**

*1. L 44. Unless one is familiar with the system, "beneath offshore arctic continental shelves" is difficult to understand.*

Response: As a response to the reviewer's comment, we changed this sentence as follows: It occurs both on land and subsea in the Arctic and subarctic regions, and underlies about 22 % of the Earth's land surface (Brown et al., 2002; Wild et al., 2022).

*2. L 47. "have developed"*

Response: Changed.

*3. L 49. 0–3 m soils = Surface 3m of soil? This sentence is a very specific comparison, but it's not clear why delta's and yedoma are included, or the reference to pre-industrial is made here. Also, is Yedoma deposit used here in reference to the Yedoma region mentioned above? The term is not introduced.*

Response: This comment refers to the sentence "Across the northern circum-polar permafrost regions, Yedoma deposits, 0–3 m soils and deltas contain about twice as much carbon as the pre-industrial atmospheric carbon pool (Hugelius et al., 2014)." The "0–3 m soils" means 0–3 m depth range in soils, or the surface 3 m of soils. We want to briefly introduce the content of organic carbon stored in circum-polar soils. We agree that this sentence is not clear. Therefore, we revised it to the following:
Permafrost regions around the world store twice as much carbon as is contained in the atmosphere at present (Hugelius et al., 2014; Friedlingstein et al., 2020). Across the northern circum-polar permafrost regions, the surface permafrost carbon pool (0–3 m depth) amounts to $1035 \pm 150$ Pg (Hugelius et al., 2014).

*4. L 70. Sakhalin peninsula and Hokkaido: are these areas in the same region? Include a short description of the geographical relation of these areas for the non-expert.*

Response: In response to the reviewer's comment, we have made a brief introduction to the relationship between these two regions (Sakhalin peninsula and Hokkaido) and the region we want to study as follows: Several studies suggested major deglacial changes in the vegetation of permafrost-affected areas during the last deglaciation, including the Lena River basin (Tesi et al., 2016), the Yukon Territory (Fritz et al., 2012), the Amur River basin (Seki et al., 2012), and Sakhalin peninsula and Hokkaido (Igarashi and Zharov, 2011), the latter two bounding the Okhotsk sea to the Northwest and North.

*5. L 88. Perhaps include that this ratio is based on microbial lipids? While the BIT index can be seen as an alternative proxy for soil-derived terrigenous OM (comparable to alkanes?) it is not quantified in this manuscript, is there a reason for this?*

Response: The BIT index in this manuscript is mentioned in this sentence "According to lignin phenols and the so-called branched and isoprenoid tetraether index (BIT), Seki et al. (2014) found that terrestrial OM from the Amur River is a major source of OM in the North Pacific Ocean at present and that terrestrial OM in surface sediments

is dominated by gymnosperms in the Okhotsk Sea." This sentence is an example to explain that lignin phenols can be used to assess vegetation development. We cited this paper by Seki et al. (2014), in which two biomarker parameters, lignin and BIT, were used. In the revised manuscript, this sentence will be deleted, and the BIT index will be included in the introduction and method sections.

*6. L 114, Alnus was mentioned in the introduction, is this the 'birch' that is referred to here? For this manuscript, it would be interesting to mention whether these vegetation changes linked to permafrost degradation or warming? A link between vegetation and drainage features linked to permafrost degradation is mentioned in the introduction, but not revisited during the description of vegetation changes here.*

Response: Birch belongs to *Betula*, and is one of the first trees to develop after the glacier retreated. *Aluns* grow in a warmer and wetter environment than birch. Therefore, pollen records indicate there were no significant changes in vegetation pattern from the LGM to about 16 ka BP, but after about 16 ka BP, birch pollen became significantly more abundant from western Alaska to the Mackenzie River (Bigelow, 2013). However, *Alnus* is a common genus in Yukon Holocene pollen records but far less common in interglacials (Schweger et al., 2011), suggesting both increasing summer temperature and moisture.

We fully agree with the reviewer that a link between vegetation changes and permafrost remobilization or climate warming should be mentioned where vegetation development is discussed in this manuscript.

*7. L 110, refer to Fig. 1 here.*

Response: It has been revised as follows: The Bering Sea is located in the north of the Pacific Ocean (Figure 1).

*8. L 121, perhaps mention that this river drains parts of Russian Siberia and northern China?*

Response: Thanks for your kind suggestion, more information about Amur River's basin has been included in the revised manuscript. See the following: The continental slope off Sakhalin Island in the Okhotsk Sea receives runoff from the Amur river, the largest river catchment in East Asia. The Amur is also one of the largest rivers in the world in terms of the annual total output of dissolved OM and substantially influences the formation of seasonal sea ice (Nakatsuka et al., 2004). The river originates in the western part of Northeast China and flows east forming the border between China and Russia.

*9. L 127. Do the authors mean "The climate of the Amur Basin is largely determined by continental patterns from Asia, as the monsoon influences the amount of precipitation from the Pacific transport to this region during the summer."? Needs to be rewritten slightly.*

Response: Thanks for your constructive suggestion, which is highly appreciated. We have revised this text and hope that it is now clearer. See the following: The climate of the Amur Basin is largely determined by continental patterns from Asia, as the

Asia monsoon influences the amount of precipitation from the Pacific transported to this region during the summer.

*10. L 131. As suggested at L. 114; Do these previous studies make a link with the permafrost collapse?*

Response: Different from the line 114 comment, we introduced the vegetation coverage in the Amur Basin at present here. The catchment of the Amur transitioned from complete permafrost coverage during the Last Glacial Maximum (LGM) to almost entirely permafrost-free conditions at present (Vandenberghe et al., 2014). Therefore, we think it might not necessary to include a link between vegetation development and permafrost collapse here. However, we will add a description of climate change from the last glacial to the deglaciation and Holocene in East Russia, and the change of vegetation in the Amur River Basin during the last interglacial period.

*11. L 214. Was this ratio calculated for this manuscript, or already published before?*

Response: The Paq ratios shown in our manuscript have been published by others. The Paq ratio of SO202-18-3/6 core was published by Meyer et al. (2019). The Paq ratio of SO178-13-6 core was published by Winterfeld et al. (2018). We also cited the Paq ratio of the XP07-C9 core (Seki et al., 2012), which is located in the Okhotsk Sea (Fig. 1) (lines 282–286, revised manuscript). The calculation of this ratio will be removed in the revised manuscript, but refer to these references when using them.

*12. L 225: 1% deactivated SiO$_2$?*

Response: Yes, it's 1% deactivated SiO$_2$. We will revise this text to address your concern.

*13. L 232. Add the reference of the manuscript where this index was introduced (and not only at L 234).*

Response: As a response to the reviewer's comment, references for all indices will be included where they first appear in this manuscript.

*14. L 234. The TEX$_{86}$ index can not be interpreted as a sea surface temperature proxy when soil input is large, as this will influence the TEX$_{86}$ ratio values directly. Often, changes in the BIT index are interpreted, as a dominant marine Thaumarchaeotal source of the isoprenoid GDGTs will be reflected in low BIT index values. The BIT index should be reported at this setting where a significant soil input is expected (it can also be used as a proxy for soil-derived organic matter, especially when interpreted coupled to concentration changes in brGDGTs and crenarchaeol). At the least, this caveat of the TEX$_{86}$ ratio should be mentioned in the text.*

Response: We fully agree with the reviewer that sea surface temperature cannot be reliably reconstructed based on the TEX$_{86}^L$ value in regions where the BIT index is high. This information will be introduced in the revised manuscript. Most BIT values in the Bering Sea are below the commonly assumed threshold value of 0.3, above which sea surface temperature reconstructions are potentially biased by terrigenous

isoGDGTs (Weijers et al., 2006). We are confident that in our study area, marine-derived GDGTs dominate over terrigenous GDGTs, suggesting that $TEX_{86}^L$ is not biased by terrigenous input. However, there are 9 samples BIT values higher than 0.3 from 13 ka BP to 10.5 ka BP, indicate that in these intervals, $TEX_{86}^L$ may not reflect the sea surface temperature change.

In order to illustrate this, BIT index values will be included in the revised manuscript. This has an added benefit, as the BIT index also reflects the increase of terrestrial organic matter input from 13 to 10.5 ka BP in the Bering Sea, which agrees with our lignin results.

*15. L 238. For the results section, please include very briefly why each parameter was reconstructed. This also allows to group the different ratios (used to reconstruct change in vegetation, vs change in degradation).*

Response: As a response to the reviewer's comment, brief descriptions of these parameters will be included as follows: The S/V and C/V ratios can be used as vegetation development, angiosperm *vs*. gymnosperm, woody tissues *vs*. non-woody tissues (Fig. 4). The 3,5Bd/V and Paq ratios can be used to indicate the change of wetland in the study area. Similar to $(Ad/Al)_s$ and $(Ad/Al)_v$ ratios, S/V, C/V, and 3,5Bd/V ratios are also affected by degradation processes (Hedges et al., 1988; Otto and Simpson, 2006).

*16. L 303. The discussion focuses on the interpretation of the ratios as vegetation markers, but only discusses the impact of provenance change afterwards. In my opinion, with this order, the reader is left to wonder whether the reconstructed vegetation changes are reliable after all. A more convincing order could be to determine the impact of sea level change on the ratios first, as sea level will dramatically impact the source and fate of the organic matter delivered to the sea floor. Here, I hypothesize that sea i) level drop = expected increase in erosion (coastal erosion, but also deeper incision of the rivers, delivering pre-aged organic matter that can/will reflect an older vegetation type, or as mentioned in the manuscript, melting of Yedoma with different S/V and C/V ratio values). Also, lower sea level (more oxic conditions), core location closer to river mouth. Then ii) sea level rise = development of anoxic conditions, better conservation. All these elements are of course mentioned in the manuscript, but not explicitly introduced as the framework in which these OM changes can be interpreted. I think this will allow to explain the shortterm changes in the vegetation ratios within the ED, BA and YD, as these seem to happen during (or rapidly following) changes in sea level. Then, the longer-term changes in the vegetation ratios can be interpreted as a change in vegetation. Following this suggestion through, the authors can discuss section 5.2. before section 5.1.*

*Instead of comparing downcore changes in vegetation and degradation proxies, perhaps a scatterplot would be more informative (plotting S/V or C/V vs 3,5Bd/V or Ad/Al ratio).*

Response: Thanks a lot for the reviewer's comment. We think sea level change influences these proxies for vegetation reconstruction and organic matter transport in our study areas in two main ways: 1) the transport time of organic matter on the shelf

or the distance between the river mouth and the study site and 2) the rate of coastal erosion.

For the first part, the change of transportation time for organic matter on the shelf can be reflected by degradation indices and radiocarbon dating. Although we don't use the radiocarbon method in this manuscript, we found no significant correlation between degradation parameters (Ad/Al) and sea level change in the two sediment cores. As mentioned in the manuscript, oxidative degradation of organic matter occurred mainly on land (Winterfeld et al., 2015). Therefore, the sea level change during the last deglaciation may have a certain impact on organic matter degradation in the Bering and Okhotsk Seas, but not much.

As the reviewer mentioned in the comment, we have discussed the impact of coastal erosion induced by sea level change on these lignin parameters. For example, in line 686–689, we mentioned coastal erosion during MWP-1B contributed to the increasing Ad/Al signals from early deglaciation to 10.5 ka BP in the Okhotsk Sea. However, the influence of coastal erosion on organic matter supplied from the different sources as reflected by increased vegetation and wetland indices is less discussed. Coastal erosion may also affect the transport of Alk and lignin from land to ocean in the Yukon and Amur Basins during the last deglaciation. We will strengthen the discussion of the effects of coastal erosion on the relative contribution of organic matter from the different sources and the transport of biomarkers. We think that sea level change can be used as an indicator for coastal erosion in this manuscript. We do, however, not think that there is a need to include a separate section in the discussion about the effect of sea level change on the biomarker parameters. However, same as general comment 2, we agree with this referee that section 5.2 should be discussed first to make this manuscript easier to understand. The structure of the discussion section has been revised based on the valuable comments of the reviewer.

We tried scatter plots before, plotting S/V or C/V vs. 3,5Bd/V or Ad/Al ratio, but we found no significant correlation between these parameters. We thus did not show these scatterplots in this manuscript.

*17. L 313. Here, the authors assume that all n-alkanes are derived from the continent, without contribution from the marine primary productivity. Is this dominant source from soil OM supported by fi bulk organic matter properties ($\delta^{13}C$)?*

Response: Thanks for the reviewer's comment. The *n*-alkanes used in this manuscript are only mid- to long-chain *n*-alkanes, but we failed to stress this in the manuscript. Combine with the second reviewer's comment, the HMW Alk of the Okhotsk Sea will be recalculated to bring it in line with that of the Bering Sea ($C_{23}$, $C_{25}$, $C_{27}$, $C_{29}$, $C_{31}$, and $C_{33}$). According to previous studies, the odd-numbered *n*-alkanes in the range of $C_{23}$ to $C_{33}$ are almost exclusively terrigenous (Eglinton and Hamilton, 1967; Otto and Simpson, 2005). Therefore, we can use the HMW Alk to reflect the contribution of terrigenous organic matter. To further quantify the contribution of marine organic matter in sediments, the $\delta^{13}C$ of bulk organic carbon and compound-specific radiocarbon analysis of lignin phenols for the two sediment cores will be analyzed in our next manuscript.

*18. L 375. Degradation, or a higher sea level?*

Response: These indices (S/V, C/V, and Paq) are influenced to some extent by degradation progresses. The distance between the river mouth and study site will increase when sea level rises, which will result in increased transport time for terrigenous organic matter on the shelf. As mentioned in the manuscript, oxidative degradation of organic matter occurs mainly on land in permafrost regions (Winterfeld et al., 2015). Therefore, the time of organic matter transport on the shelf may have little effect on its degradation, which means the impact of sea-level rise on these parameters is limited. In addition, if these parameters are mainly affected by sea level change, then they should be maximized in the B/A, not in the PB. We will include what the effects of sea level rise would be on these parameters before this conclusion in the revised manuscript.

*19. L 474. In general, I miss the impact of the distance to the river mouth in this part of the discussion. Can the authors constrain this, i.e. how much closer was the river mouth during low sea level stands, based for instance on what is known from sea level rise and current ocean floor bathymetry?*

Response: Yes, the distance of our study sites to the river mouth changed between the early deglaciation and the present, more so in the Bering than in the Okhotsk Sea. We will discuss the sea-level change for the Bering Sea from 20 ka BP to the present (Manley, 2002) in the revised manuscript, but we are not aware of a published local reconstruction of sea level change for the Okhotsk Sea during the same time. However, we do not think that this change in this distance exerts a strong control on the lignin proxies. Since the previous study suggests that the oxidative degradation of organic matter occurred mainly on land (Winterfeld et al., 2015). Thanks to the reviewer's comments, we will include the discussion of the effects of sea level change on these parameters in the revised manuscript (see comments 16 and 18).

*20. L 483. Mention those values here in-line.*

Response: Done; sentence changed to "….relatively high S/V and C/V ratios ranging from 0.47 to 1.01, and from 0.03 to 0.82, respectively (Tesi et al., 2014), indicating….". These values were also shown in Fig. 5 and 6.

*21. L 507, 508. Very interesting and important observation!*

Response: Thank you for your recognition.

*22. L 912. Not sure if this exists, but are more local reconstructions of sea-level change available? Does rebound play a role here, possibly causing a mismatch between the global sea level rise and the local conditions?*

Response: The data on global sea level change comes from Lambeck et al. (2014), which reflects global mean sea level change during the last deglaciation. We will include the local reconstruction of sea-level change for the Bering Sea from 20 ka BP (Manley, 2002) to present in the revised manuscript. We found the rate of sea level change in the Bering Sea is slower than the global average rate. However, we are not aware of a published local reconstruction of sea level change for the Okhotsk Sea during the same time. The shelf of the Okhotsk Sea is narrower than that of the Bering

Sea, so the effect of sea level change on the Okhotsk shelf may not be as strong as the Bering shelf.

We sincerely hope that these responses have addressed all your comments and suggestions. We really appreciate your efforts in reviewing our manuscript during this unprecedented and challenging time. Your careful review has helped to make our study clearer and more comprehensive.

**Reference:**

[revised manuscript text omitted]

---

## Author Comment (AC3)

Dear reviewer,

Re: Manuscript ID: cp-2022-67 and Title: Deglacial records of terrigenous organic matter accumulation off the Yukon and Amur rivers based on lignin phenols and long-chain *n*-alkanes. by Mengli Cao et al., Clim. Past Discuss., https://doi.org/10.5194/cp-2022-67-RC1, 2022

We thank you for your encouraging and helpful comments concerning our manuscript. We have studied these comments carefully and will modify the manuscript according to them. Please find the following detailed responses (blue) to these comments and suggestions.

**Major comments:**

*1. Throughout the introduction there is frequent discussion of long chain n-alkanes and n-alkyl lipids. However, when reading through to the results section it becomes clear that the authors did not extract or analyse n-alkanes themselves, rather just used published data from other studies. However, the authors do seem to have analysed isoGDGT lipids, which are not mentioned in the introduction at all. I strongly recommend re-writing the introduction to ensure that there is not a surprise for the reader when they reach the Methods section.*

Response:
We agree with the reviewer's comment regarding the introduction of our manuscript and realize the need to rewrite the introduction and include additional aspects. In particular, we included an explanation of why next to the lignin analyses we considered new and published records of biomarker-based SST and sea-ice reconstructions, and we detail in the introduction and in the methods section which of the records were newly obtained by us.

*2. Line 215: "From the polar fractions of the lipid extracts used by Meyer et al"*
- *It is not clear whether the authors have re-analysed lipids extracted by Meyer, or re-extracted their sediments. The Meyer paper reports n-alkane concentrations, but the paper mentions adding a $C_{46}$ GDGT at the time of extraction. Did Meyer pre-emptively include a $C_{46}$ GDGT? This needs clarifying, since there seems to be a contradiction.*
- *If the authors re-extracted sediments, how were they stored until this work? If they analysed lipids that had been extracted previously, how were the extracts stored?*

Response:
We realize that our initial description was confusing and re-wrote the methods section to make clear to the reader how and when the analyses were carried out and which methods were used.
- The GDGTs were extracted and analyzed by Vera D. Meyer. The internal standard of GDGTs ($C_{46}$-GDGT) was put into a vessel together with dry sediment and extract solution (dichloromethane:methanol = 9:1 (vol/vol)). After extraction, neutral compounds (including GDGTs) were extracted with *n*-hexane, and the $C_{46}$-GDGT will also be extracted with *n*-hexane. We thank the reviewer for this

comment, the description of this paragraph has been revised (lines 287–301, revised manuscript).
- The extraction of *n*-alkanes and GDGTs was carried out together. The data on *n*-alkanes were published by Meyer et al. (2019).

**Minor comments:**

*1. Line 55-56: "Around 70 % of the Yedoma region thawed beneath thermokarst lakes and streams since 14.7 ka BP"*
- *It is not clear what is meant here. 70% of the area thawed, or 70% of the area below lakes thawed?*
- *I suggest rephrasing for clarity*

Response:
- It's the thawing of 70% of the Yedoma regions, including deep thermokarst lakes (50%–60%) and streams and rivers (~ 10%–20%) (Walter Anthony et al., 2014).
- The term Yedoma is not used later, so this sentence will be removed from the revised manuscript.

*2. Lines 64, 116: "Alnus" "Populus-Salix"*
- *For readability by non-experts, it would be useful to include the common names*

Response:
- The common names of these plants will be included in the revised manuscript, such as *Alnus* (alder), *Populus -Salix* (cottonwood-willow).

*3. Lines 104-106: "The Yukon Basin was mostly unglaciated during the LGM, featuring permafrost and remains mostly so until today."*
- *This is an awkward phrasing, that could be clearer. Do you mean "remains mostly so", implying that the basin is still stable, or "remained so until today", implying that the basin has recently started to change?*

Response:
- Thanks for your kind suggestion, this sentence will be changed as follows: The Yukon Basin was mostly unglaciated during the LGM, featuring permafrost (Schirrmeister et al., 2013). Although some permafrost in the Yukon Basin thawed during the last deglaciation (Meyer et al., 2019; Wang et al., 2021), most of it is still covered by permafrost today (Fig. 1).

*4. Lines 106-107: "Arctic coasts today often are eroded at high rates of up to several meters per year"*
- *This is an awkward sentence that could be rephrased*

Response:
- This sentence will be revised as follows: The rate of Arctic coastal erosion is rapid today, the average rate of erosion for the arctic coasts is 0.5 m year$^{-1}$ (Lantuit et al., 2012; Irrgang et al., 2022), …

*5. Line 108: "… suggesting…"*

- *It is not immediately clear how the first part of this sentence suggests the second part.*

Response:
- This comment refers to the sentence "Arctic coasts today often are eroded at high rates of up to several meters per year (Lantuit et al., 2012; Couture et al., 2018), suggesting that during past periods of sea-level rise, similar or even stronger erosive forces were at play supplying vast amounts of terrigenous materials to marine sediments." 34% of the coast in the world today is affected by permafrost which is enriched in organic carbon (Hugelius et al., 2014; Schuur et al., 2015). For example, the total flux of organic carbon induced by coastal erosion for the entire Yukon coast (282 km) is 0.036 Tg C year$^{-1}$ (Couture et al., 2018). A high rate of sea level rise occurred at the start of the B/A period, and the global average rise occurs at a rate of ~ 40 mm year$^{-1}$ (Lambeck et al., 2014; Fig. 2, b). Therefore, the contribution of terrigenous organic matter in the marine shelf sediments during past periods of sea-level rise must be very high. We agree with the reviewer that this sentence is not clear, we will change it as follows: The rate of Arctic coastal erosion is rapid today, the average rate of erosion for the arctic coasts is 0.5 m year$^{-1}$ (Lantuit et al., 2012; Irrgang et al., 2022), but it may be slower than the coastal erosion in the B/A and PB periods as reflected by sea level change rate (Lambeck et al., 2014; Fig. 2, b). Coastal erosion causes a large amount of terrigenous organic matter to enter the ocean (Couture et al., 2018; Winterfeld et al., 2018), suggesting that during past periods of sea-level rise, similar or even stronger erosive forces were at play supplying vast amounts of terrigenous materials to marine sediments.

*6. Line 199: "8 lignin phenols"*
- *The eight phenols are implied in the section above, but not stated explicitly*

Response:
- The eight lignin phenols are introduced in lines 230–248. They can be classified into three groups, vanillyl phenols, syringyl phenols, and cinnamyl phenols. Vanillyl phenols consisting of vanillin (Vl), acetovanillone (Vn) and vanillic acid (Vd). Syringyl phenols, comprising syringealdehyde (Sl), acetosyringone (Sn) and syringic acid (Sd). Cinnamyl phenols that include *p*-coumaric acid (*p*-Cd) and ferulic acid (Fd). Therefore, no changes will be made in the revised manuscript about this comment.

*7. Lines 208-210: Choice of HMW n-alkanes*
- *Each paper uses different n-alkanes to calculate "HMW Alk". Is it possible to return to the original source data and recalculate so that identical alkanes are used for each basin?*

Response:
- Yes, we will recalculate the HMW Alk of the Okhotsk Sea to bring it in line with that of the Bering Sea ($C_{23}$, $C_{25}$, $C_{27}$, $C_{29}$, $C_{31}$, and $C_{33}$).

*8. Line 261: "S/V and C/V ratios"*
- *It would be useful to define the various lignin ratios in one place, and explain their uses, before starting to apply them*

Response:
- As a response to the reviewer's comment, brief descriptions of these parameters have been included in the revised results section, in agreement with the first reviewer's comment.

*9. Lines 295-296: "The deglacial evolution of the TEX$_{86}$-derived SST ranging from 4.48 to 10.8℃"*
- *This sentence needs re-writing*

Response:
- It will be revised as follows: The deglacial evolution of the TEX$_{86}^{L}$-derived SST ranging from 4.5 to 10.8 ℃.

*10. Line 301: "progressively"*
- *"Progressive" does not seem to adequately describe the data shown in the figure. A different adjective would be useful here. A clarification stating the duration over which this temperature drop happens would be helpful*

Response:
- Thanks a lot for the reviewer's comment, and this sentence will be changed as follows: The SST decreased slowly by 1℃ from 10.5 to 9.0 ka BP.

*11. Line 381: "PB,"*
- *Comma not required here*

Response:
- Changed.

We sincerely hope that these responses have addressed all your comments and suggestions. We really appreciate your efforts in reviewing our manuscript during this unprecedented and challenging time. Your careful review has helped to make our study clearer and more comprehensive.

**References:**

Couture, N. J., Irragang, A., Pollard, W., Lantuit, H., and Fritzs, M.: Coastal erosion of permafrost soils along the Yukon Coastal Plain and fluxes of organic carbon to the Canadian Beaufort Sea, J. Geophys. Res.-Biogeo., 123, 406–422, https://doi.org/10.1002/2017JG004166, 2018.

Hugelius, G., Strauss, J., Zubrzycki, S., Harden, J. W., Schuur, E. A. G., Ping, C.-L., Schirrmeister, L., Grosse, G., Michaelson, G. J., Koven, C. D., O'Donnell, J. A., Elberling, B., Mishra, U., Camill, P., Yu, Z., Palmtag, J., and Kuhry, P.: Estimated stocks of circumpolar permafrost carbon with quantified uncertainty ranges and identified data gaps, Biogeosciences, 11, 6573–6593, https://doi.org/10.5194/bg-11-6573-2014, 2014.

Irrgang, A.M., Bendixen, M., Farquharson, L.M., Baranskaya, A.V., Erikson, L.H., Gibbs, A.E., Ogorodov, S.A., Overduin, P.P., Lantuit, H., Grigoriev, M.N., Jones, B.M.: Drivers, dynamics and impacts of changing Arctic coasts, Nat. Rev. Earth Environ., 3, 39–54, https://doi.org/10.1038/s43017-021-00232-1, 2022.

Lantuit, H., Overduin, P.P., Couture, N., Wetterich, S., Aré, F., Atkinson, D., Brown, J., Cherkashov, G., Drozdov, D., Forbes, D.L., Graves-Gaylord, A., Grigoriev, M., Hubberten, H.-W., Jordan, J., Jorgenson, T., Ødegård, R. S., Ogorodov, S., Pollard, W. H., Rachold, V., Sedenko, S., Solomon, S., Steenhuisen, F., Streletskaya, I., and Vasiliev, A.: The Arctic Coastal Dynamics Database: A

New Classification Scheme and Statistics on Arctic Permafrost Coastlines, Estuar. Coasts, 35, 383–400, https://doi.org/10.1007/s12237-010-9362-6, 2012.

Lambeck, K., Rouby, H., Purcell, A., Sun, Y., and Sambridge, M.: Sea level and global ice volumes from the Last Glacial Maximum to the Holocene, P. Natl. Acad. Sci. USA, 11, 15296–15303, https://doi.org/10.1073/pnas.1411762111, 2014.

Meyer, V. D., Hefter, J., Köhler, P., Tiedemann, R., Gersonde, R., Wacker, L., and Mollenhauer, G.: Permafrost-carbon mobilization in Beringia caused by deglacial meltwater runoff, sea-level rise and warming, Environ. Res. Lett., 14, 085003, https://doi.org/10.1088/1748-9326/ab2653, 2019.

Schirrmeister, L., Froese, D., Tumskoy, V., Grosse, G., and Wetterich, S.: Yedoma: Late Pleistocene ice-rich syngenetic permafrost of Beringia, in: Encyclopedia of Quaternary Science (second edition), edited by: Elias, S. A. and Mock, C. J., Elsevier., Amsterdam, 542– 552, https://doi.org/10.1016/B978-0-444-53643-3.00106-0, 2013.

Schuur, E. A. G., McGuire, A. D., Schädel, C., Grosse, G., Harden, J. W., Hayes, D. J., Hugelius, G., Koven, C. D., Kuhry, P., Lawrence, D. M., Natali, S. M., Olefeld, D., Romanovsky, V. E., Schaefer, K., Turetsky, M. R., Treat, C. C., and Vonk, J. E.: Climate change and the permafrost carbon feedback, Nature, 520, 171–179, https://doi.org/10.1038/nature14338, 835 2015.

Walter Anthony, K. M., Zimov, S. A., Grosse, G., Jones, M. C., Anthony, P. M., Chapin III, F. S., Finlay, J. C., Mack, M. C., Davydov, S., Frenzel, P., and Frolking, S.: A shift of thermokarst lakes from carbon sources to sinks during the Holocene epoch, Nature, 511, 452–456, https://doi.org/10.1038/nature13560, 2014.

Wang, R., Kuhn, G., Gong, X., Biskabrn, B. K., Gersonde, R., Lembke-Jene, L., Lohmann, G., Tiedemann, R., and Diekmann, B.: Deglacial land-ocean linkages at the Alaskan continental margin in the Bering Sea, Front. Earth Sci., 9:712415, https://doi.org/10.3389/feart.2021.712415, 2021.

Winterfeld, M., Mollenhauer, G., Dummann, W., Köhler, P., Lembke-Jene, L., Meyer, V. D., Hefter, J., Mclntyre, C., Wacker, L., Kokfelt, U., and Tiedemann, R.: Deglacial mobilization of pre-aged terrestrial carbon from degrading permafrost, Nature Commun., 9, 3666, https://doi.org/10.1038/s41467-018-06080-w, 2018.

---

## Author Comment (AC4)

Dear reviewer,

Re: Manuscript ID: cp-2022-67 and Title: Deglacial records of terrigenous organic matter accumulation off the Yukon and Amur rivers based on lignin phenols and long-chain *n*-alkanes. by Mengli Cao et al., Clim. Past Discuss., https://doi.org/10.5194/cp-2022-67-RC1, 2022

We thank you for taking your time to review this manuscript. We really appreciate all your generous comments and suggestions. According to your advice, we amended the relevant part in the manuscript. All of your questions were answered one by one.

**General comments:**

*1. L38: It is worth stating in the abstract what is the consequence of "both types of terrestrial biomarkers [being] delivered by the same transport pathway." This contrasts with the modern river system; introducing this contrast (and a short sentence on proposed reasons why) would fit nicely at the end of the abstract.*

Response: Thanks for your kind suggestion. We found that lignin phenols and *n*-alkanes are delivered by the same transport pathway under conditions of rapid sea-level rise and shelf flooding which contrasts with the modern Arctic river system. This is one of the important conclusions in our manuscript, but we failed to stress this in the abstract. This will be revised as follows: In the modern Arctic river system, lignin and *n*-alkanes are transported from land to the ocean via different pathways, surface runoff *vs*. coastal erosion. However, accumulation rates of lignin phenols and lipids are similar in our records, suggesting that under conditions of rapid sea-level rise and shelf flooding, both types of terrestrial biomarkers are delivered by the same transport pathway. This finding suggests that the fate of terrigenous organic matter in the Arctic differs both on temporal and spatial scales.

*2. L51: Add a comma after "Holocene"*

Response: Changed.

*3. L76: Of course, the "predominance of the odd carbon number homologues" in n-alkyl lipids is only true for alkanes---alcohols and alkanoic acids exhibit the opposite preference!*

Response: As a response to the reviewer's comment, we will change this sentence as follows: Long-chain Alk with a strong predominance of the odd carbon number homologues, as well as even-numbered long-chain *n*-alkanoic acids, derive from the epicuticular waxes of vascular and aquatic plants (Eglinton and Hamilton, 1967).

*4. L79-85: The authors should clarify that this discussion on the difference in transport pathways between alkanes and lignin refers specifically to the modern systems---which they contrast with their own results in the discussion.*

Response: We agree with the reviewer's comment. This sentence will be revised as follows: Previous studies found that the delivery of lignin from land to the ocean is mainly controlled by surface discharge in modern Arctic river systems (Feng et al.,

2013) and has the potential to provide information on surface runoff processes and wetland extent (Tesi et al., 2016; Feng et al., 2015). In contrast, the long-chain $n$-alkanes (Alk) likely trace terrigenous organic matter which has been mobilized from thawing permafrost deposits in modern Arctic river systems (Feng et al., 2013) and may be transported into the marine sediment primarily following coastal erosion during shelf flooding (Winterfeld et al., 2018).

*5. L87: I second reviewer 1's opinion that the BIT index should be removed here. As far as I can tell, it is never mentioned again and, given the huge complexity and uncertainty in using this as a terrestrial OC source indicator, this brief mention only raises more questions than it answers.*

Response: Same as the first reviewer, we fully agree with them that sea surface temperature cannot be reliably reconstructed based on the $TEX_{86}^L$ value in regions where the BIT index is high. The BIT index will be included in the revised manuscript and the BIT values of core SO202-18/3/6 will be included in revised figure.

*6. Sec. 1: Overall, I agree with both previous reviewers (esp. articulated by reviewer 2) that the introduction should be reworded to clearly articulate what is new to this study and what is derived from the literature. This should also mention all of the ratios and metrics that will be used throughout this study, and briefly state their interpretation (e.g., Paq is currently not introduced as a proxy for wetland expansion until Sec. 3.2, and the interpretation of various lignin ratios is currently not clear until the discussion!)*

Response: We thank the reviewer for this constructive suggestion. We will specify which data have been published and which data are from our study in the revised manuscript. In the revised introduction, we will re-organize the introduction and believe that it will better explain why we use lignin and $n$-alkanes in this study and (biomarker-based) reconstructions of sea-surface temperatures and sea-ice extent are needed to obtain a better understanding of the deglacial changes in permafrost stability and vegetation development in the region. We will also include more detailed descriptions of the proxies used and what they can indicate ($TEX_{86}^L$, BIT, S/V, C/V, and so on), but we believe that the right place for these detailed descriptions is the methods section.

*7. L105: LGM not yet defined.*

Response: Thanks for spotting, LGM will be defined in the revised manuscript.

*8. L187-190: The inclusion of "some other oxidation products that do not necessarily originate from lignin" appears rather out-of-the-blue here, but these compounds are discussed later on. I therefore suggest at least mentioning these compounds and their utility in the introduction so that a reader knows to expect them.*

Response: Thanks a lot for the reviewer's comment, the utility of these compounds will be introduced in the revised manuscript: Unlike lignin-derived phenols (V, S, and C), 3,5Bd is absent in plant tissues, but most enriched in peat (Goñi et al., 2000b; Amon et al., 2012). The 3,5Bd/V ratio can be used as a tracer for wetland extent and to determine the degree of degradation for terrigenous organic matter.

*9. L202-205: I strongly suggest re-writing these "equations" to be proper equations, not just words written in pseudo-equation form. Something like:*
*"...calculated as follows:*
*MAR = SR×ρ,                                                                   (Eq. 1)*
*where MAR is the mass accumulation rate in g cm$^{-2}$ a$^{-1}$ , SR is the sedimentation rate in cm a$^{-1}$ , and r is the dry bulk density in g cm$^{-3}$." Etc. This would greatly simplify the reader's ability to interpret these calculations. Further down, the Paq and TEX "equations" should be restructured similarly.*

Response: As a response to the reviewer's comment, the equations in this manuscript will be changed as follows:

MAR=SR × ρ,                                                                        (Eq. 1)
MAR-lignin=MAR × Σ8 ÷ 100                                               (Eq. 2)
where MAR is the mass accumulation rate in g cm$^{-2}$ a$^{-1}$ , SR is the sedimentation rate in cm a$^{-1}$ , and ρ is the dry bulk density in g cm$^{-3}$. MAR-lignin is the mass accumulation rate of lignin (μg cm$^{-2}$ a$^{-1}$). Σ8 represents the content of the 8 lignin phenols in mg 10g$^{-1}$ dry sediment.

TEX$_{86}^{L}$ = log (GDGT-2 / (GDGT-1+GDGT-2+GDGT-3))                (Eq. 3)
SST = 27.2 × TEX$_{86}^{L}$ + 21.8                                              (Eq. 4)
The GDGT-1, GDGT-2, and GDGT-3 isoprenoid tetraether lipids with 1, 2, and 3 cyclopentane rings, which were detected by a single quadrupole mass spectrometer. The MS detector was set for selected-ion monitoring of the following $(M + H)^{+}$ ions: m/z 1300.3 (GDGT-1), 1298.3 (GDGT-2), 1296.3 (GDGT-3) (Meyer et al., 2016). SST is the sea surface temperature in °C.

According to the first reviewer's comment, the equation for Paq will be deleted.

*10. L212: Again, I would mention Paq in the introduction since this comes a bit out-of-the-blue here. This would all be clarified with a re-write of the introduction to more clearly articulate what is new and what is taken from the literature.*

Response: The Paq values used in this manuscript have been published by others (Seki et al., 2012; Winterfeld et al., 2018; Meyer et al., 2019). The Paq equation will be removed and the introduction of Paq will be included in the revised manuscript.

*11. L215: I concur with the comments of reviewer 2 for this section.*

Response: The GDGTs were extracted and analyzed by Vera D. Meyer and were contained in the polar fraction from the same total lipid extract which was used to obtain the published *n*-alkane data. GDGT analyses were performed at the same time as the *n*-alkane analyses, except that the GDGT data remained unpublished until now. We will clarify this in the revised manuscript by stating:
We further report here the relative abundances of isoprenoid glycerol dialkyl glycerol tetraether lipids (isoGDGTs). These data were obtained together with; and from the same total lipid extracts that were used for; *n*-alkane data published by Meyer et al., (2019; detailed methods therein). In brief, the internal standard of GDGTs (C$_{46}$-GDGT) was added to known amounts of dry sediment, and total lipid extracts were

obtained by ultrasonication with (dichloromethane:methanol = 9:1 (vol/vol), 3 times). After extraction and saponification, neutral compounds (including GDGTs) were recovered with $n$-hexane.

*12. L250: Should this be "11.3 ka"?*

Response: It's 11 ka BP, thanks for spotting. Changed.

*13. L275-283: It's not clear to the reader at this point what the "3,5Bd/V" ratio represents or why it is important. As mentioned above, these types of ratios should be first introduced (including their utility and interpretation) in the introduction.*

Response: The information for TEX$_{86}$$^L$, BIT, S/V, C/V, and Ad/Al will be included in the revised manuscript. Before introducing the results of these ratios, we included a brief introduction of them as follows: The S/V and C/V ratios can be used as proxies for vegetation development, angiosperm $vs$. gymnosperm, woody tissues $vs$. non-woody tissues, respectively. The 3,5Bd/V and Paq ratios can be used to indicate the change of wetland in the study area. Similar to (Ad/Al)$_s$ and (Ad/Al)$_v$ ratios, S/V, C/V, and 3,5Bd/V ratios are also affected by degradation processes (Hedges et al., 1988; Otto and Simpson, 2006).

*14. L284-294: My above comment also applies for Ad/Al ratios. These should be mentioned earlier so the reader knows how to interpret, e.g., a range of "0.19 to 0.80".*

Response: Thanks for your kind suggestion, this will be added to the revised manuscript.

*15. L296: Are TEX$_{86}$-derived SSTs really reliable to one 100th of a degree? I suggest adjusting reported precision and honestly reporting TEX$_{86}$ uncertainty here.*

Response: The reviewer is correct; we changed our reporting of TEX$_{86}$$^L$-derived SST estimates to only one decimal digit ("…SST ranging from 4.5 to 10.8°C").

*16. Sec. 5.1-5.2: I concur with reviewer 1's suggestion to restructure these sections to begin with a discussion on terrestrial OM sources, fluxes, accumulation rates, etc. and then move into an interpretation of these biomarker ratios, including potential impacts of degradation.*

Response: We thank the reviewer for this constructive suggestion. We will change the structure of the discussion section, discussing organic matter sources based on biomarker concentrations first, followed by vegetation development in the two basins.

*17. L555-556: This is an interesting finding, but it is only mentioned in passing here. Why do the authors think these delivery mechanisms have changed relative to the modern? I would appreciate a bit more discussion on this topic, as I think some reviewers will find it highly relevant.*

Response: By comparing the mass accumulation rates of lignin and $n$-alkanes and the relationship between sea level change and mass accumulation rate, we found lipids

and lignin might have been delivered to the ocean by identical processes, i.e., runoff and erosion. For example, if the transport of $n$-alkanes were mainly affected by sea level change, the mass accumulation rate of $n$-alkanes should be maximized in the B/A, not in the PB in the two sediment cores (Fig. 2, 3). However, further analysis of whether there is a difference in the mode of transport for the two biomarkers requires compound-specific radiocarbon analysis, where systematically different ages of the two types of compounds would be indicative of different transport and supply mechanisms. We will address this question by radiocarbon dating of lignin phenols in a follow-up project, which is not the subject of this manuscript. Here, we focus on the vegetation development in the two river basins.

We agree with the reviewer that this finding requires more elaboration. We will emphasize this (and the difference between our results and previous findings) in the revised introduction and discussion.

First, we will introduce previous studies on the transport of lignin and $n$-alkanes in the modern Arctic river systems in the revised introduction. Second, we will introduce the research objectives of this study: Previous studies have reconstructed the mobilization of terrigenous organic matter from degrading permafrost in the Okhotsk (Winterfeld et al., 2018) and Bering shelves (Meyer et al., 2019) during the last deglaciation based on long-chain $n$-alkyl lipids results. However, no records exist that combine lignin and Alk data to explore the potentially different transport of terrestrial organic matter archived in them during the last deglaciation.

Third, we will emphasize this point in the discussion, for example, "In the B/A, all biomarker fluxes increased and reached short maxima (Fig. 2c, d). The rate of sea level rise also reached its maximum since the LGM. If Alk had been transported to the ocean primarily through erosion of deep permafrost deposits, as has been suggested for the modern Arctic river transport systems (Feng et al., 2013), then Alk MAR would have been at its maximum.", and "The rate of sea level change was lower during the PB than that in the B/A, but the MARs of Alk and lignin reached their maxima, and the discharge of Yukon River also increased from the B/A to the PB. Therefore, both coastal erosion and surface runoff may affect the transport of Alk and lignin from land to ocean in the Yukon Basin during the last deglaciation.". In the discussion of Amur Basin for terrigenous organic matter mobilization during the last deglaciation, the transport of lignin and $n$-alkanes will also be discussed in a similar manner.

We sincerely hope that these responses have addressed all your comments and suggestions. We really appreciate your efforts in reviewing our manuscript during this unprecedented and challenging time. Your careful review has helped to make our study clearer and more comprehensive.

**Reference:**

Eglinton, G. and Hamilton, R.J.: Leaf epicuticular waxes-The waxy outer surfaces of most plants display a wide diversity of fine structure and chemical constituents, Science, 156, 1322–35, 1967.

Feng, X., Vonk, J. E., van Dongend, B. E., Gustafssone, Ö., Semiletov, I. P., Dudarev, O. V., Wang, Z., Montlucon, D. B., Wacker, L., and Eglinton, T. I.: Differential mobilization of terrestrial carbon pools in Eurasian Arctic river basins, P. Natl. Acad. Sci. USA, 110, 14168–14173, https://doi.org/10.1073/pnas.1307031110, 2013.

Feng, X., Gustafssone, Ö., Holmes, R.M., Vonk, J. E., van Dongend, B. E., Semiletov, I. P., Dudarev, O. V., Yunker, M. B., Macdonald, R. W., Montlucon, D. B., and Eglinton, T. I.: Multi-molecular tracers of terrestrial carbon transfer across the pan-Arctic: comparison of hydrolyzable

components with plant wax lipids and lignin phenols, Biogeosciences, 12, 4841–4860, https://doi.org/10.5194/bg-12-4841-2015, 2015.

Meyer, V. D., Max, L., Hefter, J., Tiedemann, R., and Mollenhauer, G.: Glacial-to-Holocene evolution of sea surface temperature and surface circulation in the subarctic northwest Pacific and the Western Bering Sea, Paleoceanography, 31, 916–927, https://doi.org/10.1002/2015PA002877, 2016.

Meyer, V. D., Hefter, J., Köhler, P., Tiedemann, R., Gersonde, R., Wacker, L., and Mollenhauer, G.: Permafrost-carbon mobilization in Beringia caused by deglacial meltwater runoff, sea-level rise and warming, Environ. Res. Lett., 14, 085003, https://doi.org/10.1088/1748-9326/ab2653, 2019.

Tesi, T., Muschitiello, F., Smittenberg, R. H., Jakobsson, M., Vonk, J. E., Hill, P., Andersson, A., Kirchner, N., Noormets, R., Dudarev, O., Semiletov, I., and Gustafsson, Ö.: Massive remobilization of permafrost carbon during post-glacial warming, Nature Commun., 7, 13653, https://doi.org/10.1038/ncomms13653, 2016.

Winterfeld, M., Mollenhauer, G., Dummann, W., Köhler, P., Lembke-Jene, L., Meyer, V. D., Hefter, J., Mclntyre, C., Wacker, L., Kokfelt, U., and Tiedemann, R.: Deglacial mobilization of pre-aged terrestrial carbon from degrading permafrost, Nature Commun., 9, 3666, https://doi.org/10.1038/s41467-018-06080-w, 2018.

---

## Author Response (AR1)

Dear Erin McClymont,

Re: Manuscript ID: cp-2022-67 and Title: Deglacial records of terrigenous organic matter accumulation off the Yukon and Amur rivers based on lignin phenols and long-chain *n*-alkanes. by Mengli Cao et al., Clim. Past Discuss., https://doi.org/10.5194/cp-2022-67-RC1, 2022

We really appreciate your efforts in reviewing our manuscript. Your careful review has helped to make our study clearer and more comprehensive. According to your suggestions and comments, we amended the relevant part in the manuscript. All of your questions are answered one by onein the following.

**Comments to the author:**
Thank you for outlining your responses to the three reviewers of your manuscript. The reviewers gave positive comments and constructive suggestions for how to ensure that readers can follow your new data interpretations. Your responses indicate that these can largely be incorporated into the manuscript.

All reviewers recommended significant revisions to the Introduction, which should be carefully undertaken before submitting a revised version of the manuscript, since this is important for ensuring the new data and analyses are clear, as well as the framework in which you will interpret the results. Likewise, significant changes to the Discussion structure have been suggested and positively responded to, but will require careful checks to ensure that there remains a good flow through the text.

Response: We agree with this comment regarding the introduction of our manuscript and realize the need to re-organize the text and include additional aspects. The structure of the discussion section has been changed in the revised manuscript. We discussed organic matter sources based on biomarker concentrations first, followed by vegetation development in the two basins.

**Some minor queries:**
- *reply to reviewer 1 comment on line 44. Your suggested response does not completely clarify the "beneath offshore Arctic continental shelves". An alternative: "It occurs both on land and on the continental shelves offshore, and underlies..."*
Response: Changed (line 45, revised manuscript).

- *reply to reviewer 2 comment on line 88. I don't think the reviewer was asking for this to be removed, but rather to flag that in contrast to the lignin phenols (which you have already explained), the BIT index could have been noted as having a different bacterial (soil) source. Is a deletion necessary?*
Response: This sentence is an example to explain that lignin phenols can be used to assess vegetation development and it comes from this paper by Seki et al. (2014). We'll cite this paper in the discussion of vegetation changes. This sentence thus will be removed to avoid duplication. In addition, the comment on line 88 is for the explanation of the parameter BIT, and the explanation of which has been added in the revised introduction. Therefore, there is no need to keep this sentence in the revised manuscript.

- *line 91-32: is it true that no records exist which combine "both types" of biomarker (which two are you referring to here? I assume n-alkanes and lignin phenols), or is it that such a record does not exist in this region? This needs clarification.*

Response: Thanks for your kind suggestion, which is highly appreciated. We have revised this text and hope that it is now clearer: "However, no records exist that combine lignin and *n*-alkanes data to explore the potentially different transport of terrestrial OM archived in Arctic marine sediments during the last deglaciation." (line 109–111, revised manuscript)

- *reply to reviewer 2 comment on line 238. The first sentence of your revised text needs to be more clear about which ratio refers to which signal e.g. what does S/V indicate, and what does C/V indicate? In this sentence you mention 2 ratios but it isn't clear which is which. An alternative: "Vegetation development can be assessed using the S/V (angiosperm vs. gymnosperm) and C/V ratios (woody tissues vs. non-woody tissues). You need to add citations of the literature to support these interpretations.*

Response: Thanks a lot for this comment. Brief descriptions of these parameters have been included in the revised results section: "Vegetation development can be assessed using the S/V (angiosperm *vs*. gymnosperm) and C/V ratios (woody tissues *vs*. non-woody tissues) (Hedges and Mann, 1979). The 3,5Bd/V and Paq ratios can be used to indicate the change of wetland extent in the study area (Goñi et al., 2000b; Amon et al., 2012). Similar to $(Ad/Al)_s$ and $(Ad/Al)_v$ ratios, S/V, C/V, and 3,5Bd/V ratios are also affected by degradation processes (Ertel and Hedges, 1985; Hedges et al., 1988; Otto and Simpson, 2006)." (line 364–369, revised manuscript)

- *reply to reviewer 2 comment about the GDGT extractions. Thank you for clarifying that the GDGT data were newly generated for this paper, alongside the n-alkane data which has previously been published. The methods as outlined here lack some detail: you note that the GDGTs were extracted from the extract using hexane, but is that from a total extract or did you do some column chromatography or saponification as clean-up steps? If not, this sounds like you took a dry extract and dissolved it in hexane and injected that onto the LC, but the mention of "neutral compounds" sounds like you saponified first, and it is also common to filter samples before running on the LC. Can you expand the method details in the revised submission? [I subsequently found that in your response to Reviewer 3 you include notes about saponification. Please ensure that your methods section is appropriately detailed for a reader to replicate your methods]*

Response: Yes, the details of this method have been included in the revised manuscript (line 292–306, revised manuscript).

- *reply to reviewer 2 comment on lines 104-106. A suggestion to change the text slightly: "The Yukon Basin was mostly unglaciated during the LGM, but had permafrost (Schirrmeister et al., 2013). Although some permafrost in the Yukon Basin thawed during the last deglaciation (Meyer et al., 2019; Wang et al., 2021), most of the basin is still covered by permafrost today (Fig. 1)."*

Response: Changed  (line 143–146, revised manuscript).

- *reply to reviewer 2 comment on lines 106-107. A suggestion: "Arctic coastal erosion is rapid today, with average rates of erosion at 0.5 m year$^{-1}$ (Lantuit et al., 2012; Irrgang et al., 2022), ..."*

Response: Changed (line 146–147, revised manuscript).

- *reply to reviewer 2 comment on line 108. Your reply gives some additional detail which is helpful, but it doesn't quite get to the issue which is why a time of sea level rise is expected to cause increased erosion. If you could break this long sentence into several smaller ones, you may be able to lead the reader through (i) high rates of coastal erosion today; (ii) expected sea level rise in the past (iii) rising sea levels cause more erosion (with supporting literature) (iv) past rises in sea level might then have contributed more coastal erosion, including organic matter. At the moment, part (iii) is missing from your interpretation.*

Response: We agree with this comment and we will change it as follows: "Arctic coastal erosion is rapid today, with average rates of erosion at 0.5 m year$^{-1}$ (Lantuit et al., 2012; Irrgang et al., 2022). Sea level rise will lead to greater wave impact on arctic shorelines which increases the coastal erosion (Lantuit et al., 2012). This suggests that during past times of rapid sea-level increase like in the B/A and PB periods coastal erosion was more intense than it is today (Lambeck et al., 2014; Fig. 2, b). Coastal erosion causes a large amount of terrigenous organic matter to enter the ocean (Couture et al., 2018; Winterfeld et al., 2018), suggesting that during past periods of sea-level rise, similar to today or even stronger erosive forces were at play supplying vast amounts of terrigenous materials to marine sediments." (line 146–154, revised manuscript).

- *reply to reviewer 2 comment on line 199. I agree with the reviewer that saying there are 8 here but not outlining which 8 until line 230 is confusing.*

Response: In the revised manuscript, the 8 lignin phenols are introduced in line 233 and following, while $\Lambda 8$ and $\Sigma 8$ are introduced in lines 267 and 274.

- *reply to reviewer 2 comment on line 295-296. I don't see a change here in your reply. A suggestion "The deglacial evolution of the TEX86L-derived SST shows an overall warming, from ~4.5 °C at (date) ka BP to 10.8 °C at (date) ka BP." Add in the relevant dates here and the sentence may be easier to follow.*

Response: It has been changed in the revised manuscript (line 356–357).

---

## Author Response (AR2)

Dear Erin McClymont,

Re: Manuscript ID: cp-2022-67 and Title: Deglacial records of terrigenous organic matter accumulation off the Yukon and Amur rivers based on lignin phenols and long-chain *n*-alkanes. by Mengli Cao et al., Clim. Past Discuss., https://doi.org/10.5194/cp-2022-67-RC1, 2022

Thanks very much for your kind work and consideration on publication of our paper. On behalf of my co-authors, we would like to express our great appreciation to you and reviewers. According to your comments, we have revised the manuscript extensively. Our responses are given in a point-by-point manner below.

**Comments to the author:**
*Thank you for addressing the reviewer and editor comments, and submitting the tracked changes document. I am happy to recommend publication of the manuscript subject to you making some final clarifications / checks:*

1. *Paragraph starting line 88 has no citations of published literature: can you insert some here to back up your statements?*

Response: Yes, I can. Some references have been included in this paragraph.

2. *Paragraph starting line 112 has only citations for the original proxy proposals, but published examples of where these proxies have been applied could be used to strengthen the statements made.*

Response: Thanks for your kind suggestion, which is highly appreciated. Two published examples have been included in this paragraph, one is for the $TEX_{86}$ temperature proxy (Meyer et al., 2016), the other is about the lignin phenols and the BIT index (Seki et al., 2014).

3. *Line 127: remove "drainages"*

Response: Changed (line 140, revised manuscript).

4. *Line 524: this doesn't make sense. Was the permafrost lost after the LGM, or has it been recently lost ("today")?*

Response: We agree with this comment and we will change it as follows: "The Amur Basin was completely covered with permafrost during the LGM (Vandenberghe et al., 2014) and almost all of the permafrost was lost until today as a result of permafrost mobilization during the last deglaciation." (line 535–537, revised manuscript)

5. *Line 573: check that "resuscitation" is the correct term here.*

Response: It has been changed to "development" in the revised manuscript (line 585, revised manuscript).

**Reference:**

Meyer, V. D., Max, L., Hefter, J., Tiedemann, R., and Mollenhauer, G.: Glacial-to-Holocene evolution of sea surface temperature and surface circulation in the subarctic northwest Pacifific and the Western Bering Sea, Paleoceanography, 31, 916–927, https://doi.org/10.1002/2015PA002877, 2016.

Seki, O., Mikami, Y., Nagao, S., Bendle, J.A., Nakatsuka, T., Kim, V. I., Shesterkin, V. P., Makinov, A. N., Fukushima, M., Mossen, H. M., and Schouten, S.: Lignin phenols and BIT index distribution in the Amur River and the Sea of Okhotsk: Implications for the source and transport of particulate terrestrial OC to the Ocean, Prog. Oceanogr., 126, 146-154, http://dx.doi.org/10.1016/j.pocean.2014.05.003, 2014.